# `PRING`: Rethinking Protein-Protein Interaction Prediction from Pairs to Graphs

**Xinzhe Zheng**[1,2*]   **Hao Du**[2*]   **Fanding Xu**[1,7*]   **Jinzhe Li**[2,4*]
**Zhiyuan Liu**[1†]   **Wenkang Wang**[1,8]   **Tao Chen**[3,4]   **Wanli Ouyang**[2,6,9]
**Stan Z. Li**[5]   **Yan Lu**[2,6†]   **Nanqing Dong**[2,3†]   **Yang Zhang**[1†]

[1]National University of Singapore   [2]Shanghai Artificial Intelligence Laboratory
[3]Shanghai Innovation Institute   [4]Fudan University   [5]Westlake University
[6]Chinese University of Hong Kong   [7]Xi'an Jiaotong University
[8]Central South University   [9]Shenzhen Loop Area Institute

## Abstract

Deep learning-based computational methods have achieved promising results in predicting protein-protein interactions (PPIs). However, existing benchmarks predominantly focus on isolated pairwise evaluations, overlooking a model's capability to reconstruct biologically meaningful PPI networks, which is crucial for biology research. To address this gap, we introduce `PRING`, the first comprehensive benchmark that evaluates **PR**otein-protein **IN**teraction prediction from a **G**raph-level perspective. `PRING` curates a high-quality, multi-species PPI network dataset comprising 21,484 proteins and 186,818 interactions, with well-designed strategies to address both data redundancy and leakage. Building on this golden-standard dataset, we establish two complementary evaluation paradigms: (1) **topology-oriented** tasks, which assess intra and cross-species PPI network construction, and (2) **function-oriented** tasks, including protein complex pathway prediction, GO module analysis, and essential protein justification. These evaluations not only reflect the model's capability to understand the network topology but also facilitate protein function annotation, biological module detection, and even disease mechanism analysis. Extensive experiments on four representative model categories, consisting of sequence similarity-based, naive sequence-based, protein language model-based, and structure-based approaches, demonstrate that current PPI models have potential limitations in recovering both structural and functional properties of PPI networks, highlighting the gap in supporting real-world biological applications. We believe `PRING` provides a reliable platform to guide the development of more effective PPI prediction models for the community. The dataset and source code of `PRING` are available at https://github.com/SophieSarceau/PRING.

## 1   Introduction

Protein-protein interactions (PPIs) refer to the physical or functional association between proteins, which is central to most biological processes, such as signal transduction [1], gene regulation [2], and immune response [3]. They also play a key role in the study of disease mechanisms [4]. For instance, the interactions between certain cellular proteins can trigger abnormal signaling pathways that promote tumor growth and metastasis in cancer [5]. By revealing those types of PPI, small molecules are developed to disrupt the interactions, thereby inducing cancer cell death [6]. Therefore, characterizing the organization of PPIs is fundamental for understanding the molecular basis of health and advancing drug discovery in precision medicine [7, 8].

---

*Equal contribution.
†Corresponding author.

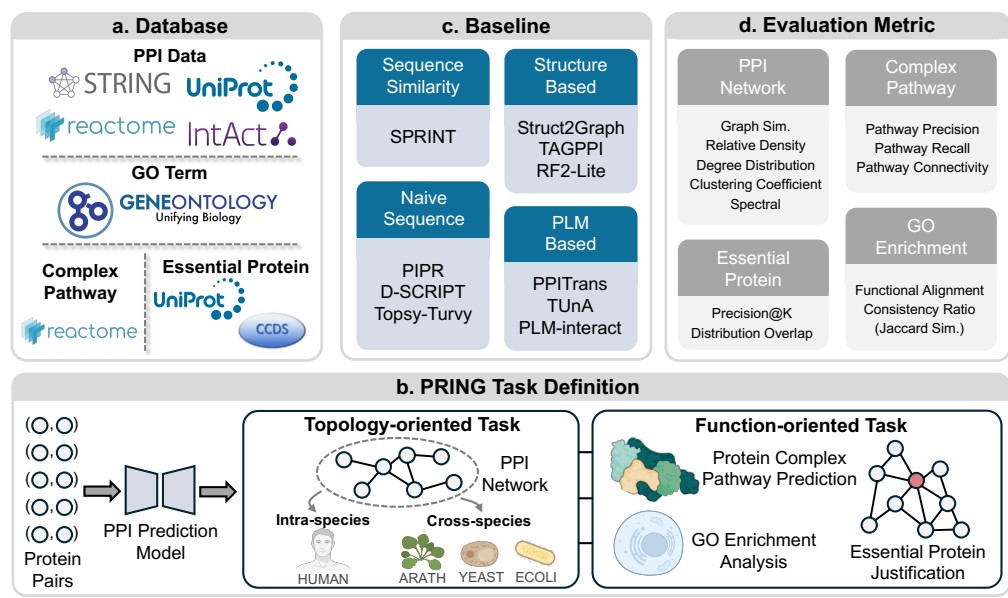

Figure 1: Overview of PRING benchmark. (**a**) Diverse databases are used to construct the PRING. (**b**) PRING includes two topology-oriented tasks and three function-oriented tasks for extensive evaluation. (**c**) List of baseline models, consisting of sequence similarity-based, naive sequence-based, structure-based, and PLM-based methods. (**d**) Evaluation metrics used for each task in the PRING.

Experimental techniques are adopted to identify PPIs in traditional biological research, including instrument-based methods such as X-ray crystallography [9, 10] and cryo-electron microscopy [11, 12], as well as high-throughput approaches like yeast two-hybrid [13, 14] and cross-linking mass spectrometry [15, 16]. Nevertheless, these methods are limited by low scalability and incomplete coverage of the interactome. The rise of deep learning provides powerful computational alternatives for PPI identification [17, 18, 19, 20], which designs neural networks to capture structural and sequence patterns of proteins to predict PPIs and demonstrates promising performance. More recently, the development of AlphaFold [21, 22, 23] and protein language models (PLMs) [24, 25, 26, 27] has led to remarkable breakthroughs in PPI prediction, enabling more accurate and large-scale identification of PPI directly from protein sequences.

Although current deep learning-based PPI prediction models achieve high pairwise accuracy on standard benchmarks [28, 29, 30, 31, 32, 17], these evaluations treat each interaction in isolation and overlook how individual predictions contribute to the cohesive PPI networks. While some studies [33, 17] have applied these models to reconstruct PPI networks in specific contexts (*e.g.*, bovine rumen), a systematic and fair evaluation of network-level performance remains absent. Since PPI networks support biological insights from both **topological** and **functional** perspectives [34, 35], it is important to evaluate how well models recover both aspects. Topologically, PPI networks exhibit properties such as sparsity and local communities, which reflect the modular organization of cellular processes. Functionally, they offer a foundation for annotating uncharacterized proteins and discovering coherent biological modules. Hence, we pose the following research question: *How well do current PPI models demonstrate their pure capabilities in recapitulating the structural and functional features of PPI networks?*

To fill this blank, we propose PRING, a multi-organisms holistic benchmark that evaluates the **PR**otein-protein **IN**teraction prediction from a **G**raph perspective. As illustrated in Fig. 1, PRING compiles high-confidence physical interactions across four organisms, Human, Arath, Ecoli, and Yeast, sourced from STRING [36], UniProt [37], Reactome [38], and IntAct [39], yielding 21,484 proteins and 186,818 interactions, with efforts made to minimize both data redundancy and leakage. Through this golden dataset, we benchmark representative PPI prediction methods, including sequence similarity-based, naive sequence-based, PLM-based models, and structure-based, across five tasks: two topology-oriented, namely intra-species and cross-species PPI network construction, to assess whether models can recover the topology properties of PPI networks, such as network density and local community structures, and facilitating cross-species biological knowledge transfer study, and three function-oriented, including protein complex pathway prediction, GO functional module

analysis, and essential protein justification, which can support disease mechanism analysis, protein function annotation, and therapeutic target identification. Collectively, these evaluations reveal whether a model is capable of capturing both the topological and the functional semantics of real interactomes, moving beyond pairwise classification to network-level understanding.

Extensive experiments yield the following insights: (1) current models tend to generate overly dense graphs, diverging from the sparsity nature of real PPI networks; (2) predicted PPI modules exhibit limited functional alignment with the ground-truth, restricting their utility in downstream tasks such as pathway reconstruction and function annotation; and (3) reconstructed graphs struggle to separate essential from non-essential proteins, indicating that critical topological signals remain uncaptured; (4) classification metrics cannot completely reflect a model's ability to recover network structure, highlighting the need for graph-level evaluations. While these findings highlight challenges, we hope they can serve as a positive step forward for the community. PRING complements existing benchmarks and provides a reliable platform for more holistic research in modeling PPI networks.

To sum up, our key contributions include:

- We propose PRING, to the best of our knowledge, the first comprehensive benchmark that evaluates PPI prediction models from topological and functional perspectives on PPI networks.

- Through extensive experiments, we demonstrate that existing models exhibit limited functional awareness and poor topological fidelity, revealing a potential gap between computational approaches and their applicability in biological research.

- We release a fully reproducible pipeline, including dataset construction and model evaluation tools, to facilitate future research and benchmarking efforts within the community.

## 2 Related Work

### 2.1 Protein-Protein Interaction Prediction Benchmark

Numerous PPI benchmarks have been developed to assess the effectiveness of prediction models. Most existing benchmarks [40, 41, 31, 30, 32] focus on physical interaction evaluations, which curate Human and Yeast PPI datasets from resources such as DIP [42], UniProt [37], and HPRD [43], applying filtering criteria to ensure data quality and evaluating models using binary classification metrics. To address the limitations of small-scale and single-species benchmarks, D-SCRIPT [17] introduced cross-species evaluation by sampling 65,138 interactions across multi-species from STRING [44, 45, 36]. Meanwhile, Bernett *et al.* [46] further raised concerns about data leakage caused by naive splitting strategies (*e.g.*, random splits), which can inflate model performance by enabling shortcut learning [47]. To mitigate this, they proposed more rigorous splitting protocols that revealed performance drops across benchmarks. Other studies [17, 33] built PPI networks to perform GO enrichment analysis or functional module detection in specific biological scenarios, such as bovine rumen in cows, to broaden the scope of model applications, but lack the comparison with ground truth. Additionally, several works [48, 49] constructed PPI benchmarks annotated with functional interaction types or binding sites, aiming to train models to predict these properties via multi-class classification. However, the aforementioned benchmarks primarily focus on pairwise interaction accuracy and lack a fair and holistic evaluation of a model's capability in PPI network construction. To address these gaps, we introduce PRING, a graph-centric benchmark that evaluates PPI models from both topological and functional perspectives.

### 2.2 Protein-Protein Interaction Prediction Model

Computational methods for PPI prediction are gaining popularity due to their efficiency and reliability in understanding complex biological systems [50, 51]. Early studies [52] often inferred unknown interactions by leveraging sequence similarity with known homologous interaction pairs. Although these methods are efficient, their accuracy is limited, especially for novel interactions. The advent of deep learning [53, 54, 55] brought a powerful computational tool for PPI prediction. Sequence-based models [32, 48, 17, 56] utilized convolutional neural networks (CNNs) [57, 58], recurrent neural networks (RNNs) [59, 60], to learn interaction patterns from raw protein sequences. More recently, protein language models (PLMs) [61, 62], pre-trained on large-scale protein datasets, have shown strong performance across biological tasks such as contact prediction and function annotation. Motivated by this, PLM-based methods [19, 63, 64] leverage rich and context-aware representations to predict PPIs, achieving state-of-the-art results on some benchmarks. Since proteins can be naturally

represented as graphs, some methods [65, 66] utilized protein structure as input and employed Graph Neural Networks (GNNs) [67, 68] to predict PPIs. Meanwhile, with the success of structure prediction tools like AlphaFold [21] and RoseTTAFold [69], structure-based methods [70, 71] have also emerged, offering improved biological explainability. Additionally, some studies [72, 73, 49] applied GNNs to infer functional interaction types based on the known PPI network as input. In this work, we evaluate PPI prediction models designed to identify physical interactions from a graph-level perspective, providing insights into their ability to reconstruct biologically meaningful interactomes.

## 2.3 Multi-Modal Learning for Enhanced Biological Representation

An emerging research direction seeks to overcome the limitations of single-modality PPI models by constructing richer representations from multimodal data. One of the approaches is to unify molecular information, such as structure and sequence, with semantic data like natural language to improve functional prediction. This paradigm is exemplified by models that connect protein data to text for better interpretability [27, 74, 75]. The approach has been broadly explored by fusing chemical structures with text [76, 77, 78, 79, 80, 81]. Such advancements in unified learning offer a promising path to developing more powerful PPI models that can address the topological and functional challenges identified by the PRING benchmark.

## 3 Protein-Protein Interaction Prediction from a Graph Perspective

In this section, we present PRING, a comprehensive and fair benchmark designed to evaluate PPI prediction models from a graph perspective. To construct such a benchmark, we first curate PPIs from diverse resources and filter redundant proteins based on sequence similarity and functional overlap to ensure data quality, and then we partition data into training and test sets with non-overlapping proteins to prevent data leakage, yielding a PPI network dataset comprising 21,484 proteins, 186,818 PPIs (Sec. 3.1). A detailed comparison between PRING and existing PPI benchmarks is shown in Tab. 1. Based on this golden dataset, we develop a set of tasks to assess model performance in capturing both the topology and biological functionality of PPI networks (Sec. 3.2).

Table 1: Comparison of PRING with existing PPI benchmarks. (Seq.Sim.: sequence similarity, N/A: not available.)

| Benchmark | Year | #Proteins | #Pairs | Seq. Sim. | Multi Species | Leakage Free | Graph Evaluations |
|---|---|---|---|---|---|---|---|
| GUO [40] | 2008 | 2,497 | 5,594 | N/A | ✗ | ✗ | ✗ |
| PAN [31] | 2010 | 9,476 | 36,630 | N/A | ✗ | ✗ | ✗ |
| HUANG [41] | 2016 | 3,163 | 3,899 | 25% | ✗ | ✗ | ✗ |
| DU [30] | 2017 | 4,424 | 17,257 | 40% | ✗ | ✗ | ✗ |
| RICHOUX [32] | 2019 | 8,333 | 45,765 | N/A | ✗ | ✗ | ✗ |
| SHS27K [48] | 2019 | 1,663 | 7,401 | 40% | ✗ | ✗ | ✗ |
| SHS148K [48] | 2019 | 5,082 | 43,397 | 40% | ✗ | ✗ | ✗ |
| D-SCRIPT [17] | 2021 | 19,086 | 65,138 | 40% | ✓ | ✗ | ✗ |
| HIPPIE [46] | 2024 | 10,819 | 137,250 | 40% | ✗ | ✓ | ✗ |
| PRING (Ours) | 2025 | 21,484 | 186,818 | 40% | ✓ | ✓ | ✓ |

proteins, 186,818 PPIs (Sec. 3.1). A detailed comparison between PRING and existing PPI benchmarks is shown in Tab. 1. Based on this golden dataset, we develop a set of tasks to assess model performance in capturing both the topology and biological functionality of PPI networks (Sec. 3.2).

### 3.1 Data Collection

Diverse, reliable, and non-redundant interaction data is essential for effective and comprehensive evaluation of PPI prediction models [46]. To achieve this goal, PRING is mainly assembled in two stages: (1) integrating high-confidence multi-species PPIs; and (2) applying homology- and function-based filters to eliminate redundancy and utilizing a leakage-free splitting protocol to prevent potential leakage. A pipeline overview is shown in Fig. 2, and detailed data documentation is given in Apx. C.

**Resource Integration.** We begin by aggregating experimentally validated PPIs from UniProt [37], Reactome [38], and IntAct [39], and complement these sources with high-confidence interactions (combined score > 0.7 [82]) from STRING [36] to ensure both data quality and comprehensive coverage, while mitigating the noise inherent in some STRING interactions. To further enhance data reliability and prepare for downstream function-oriented analyses, we retain only annotated proteins with known functions in SwissProt [83]. Additionally, four phylogenetically distinct organisms, including Human, Arabidopsis thaliana (Arath), Escherichia coli (Ecoli), and Yeast, are selected using NCBI Taxonomy identifiers [84] to match with the target species and increase the species diversity. This protein filter step not only considers data quality but also emphasizes data diversity and coverage, providing a critical foundation for constructing our gold-standard dataset.

**Minimal Redundancy and Leakage.** Our primary objective is to build a gold-standard dataset with minimal data redundancy and leakage for fair evaluations of PPI models. To this end, we first perform sequence clustering using MMSeqs2 [85, 86] and retain only protein pairs with sequence identity ≤ 40%, following prior studies [17, 30]. Meanwhile, we remove proteins that share identical function

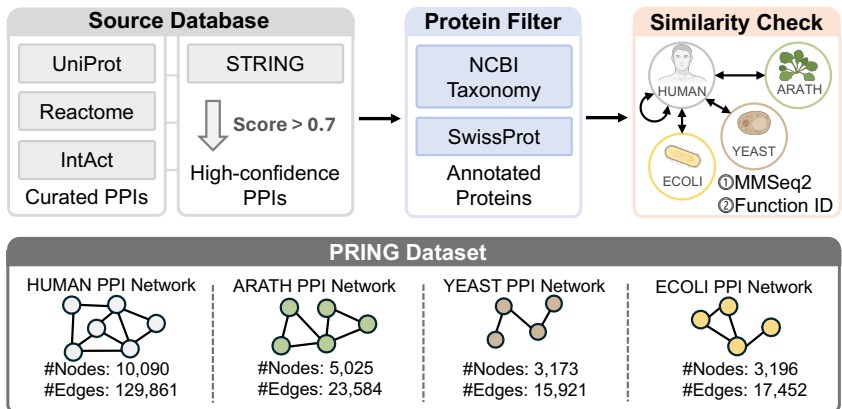

Figure 2: Data collection pipeline for the PRING. PPIs are first curated from comprehensive databases. Proteins are then filtered and mapped using SwissProt and NCBI Taxonomy to target species. Redundant interactions are removed through sequence and functional similarity checks to ensure data quality. The resulting PPI networks include four species: Human, Arath, Yeast, and Ecoli.

IDs across species as an additional safeguard to reduce data redundancy further. Additionally, a leakage-free protocol [46] is used to split the dataset with non-overlapping proteins, further preventing data leakage in model learning. The above filtering steps result in high-quality PPI networks for each organism in the PRING dataset. Specifically, the Human network comprises 10,090 proteins and 129,861 PPIs; Arath includes 5,025 proteins and 23,584 PPIs; Ecoli contains 3,196 proteins and 17,452 PPIs; and Yeast consists of 3,173 proteins and 15,921 PPIs. We believe this dataset provides a high-confidence resource for PPI benchmarking and can facilitate fair and rigorous comparisons among different PPI models.

## 3.2 Task Suite

Through the PRING dataset, we develop five tasks to systematically evaluate how well PPI prediction models recapitulate both the **structural topology** and **functional properties** of PPI networks.

### 3.2.1 Topology-Oriented Task

To evaluate whether a PPI prediction model can faithfully recover the topology of interaction networks, we first formulate the PPI network reconstruction task as follows. Given a PPI graph with $N$ proteins, each protein is represented by its features $x_i = \{s_i, c_i\}, \quad i = 1, \ldots, N$, where $s_i$ is the amino-acid sequence and $c_i$ is any auxiliary context. The model $f$ predicts an interaction label for each protein pair $(i, j)$ with $1 \leq i < j \leq N$, according to: $I_{i,j}^{\mathrm{pred}} = f(x_i, x_j) \in \{0, 1\}$. Here, $I_{i,j}^{\mathrm{pred}} = 1$ indicates a predicted interaction between protein $i$ and protein $j$, while $I_{i,j}^{\mathrm{pred}} = 0$ denotes no predicted interaction. We then aggregate all positive predictions to form the reconstructed network:

$$G^{\mathrm{pred}} = \mathrm{Aggregate}\big\{(i, j) \mid I_{i,j}^{\mathrm{pred}} = 1, \ 1 \leq i < j \leq N\big\}. \tag{1}$$

Each reconstructed graph $G^{\mathrm{pred}}$ is then compared against its corresponding ground-truth graph $G^{\mathrm{true}}$. To evaluate the model's ability of identifying topology properties like network density and cross-species knowledge transfer, we introduce two tasks: (1) intra-species network construction, where both training and test are performed on the same species; and (2) cross-species network construction, where the model is trained on one species and evaluated on another.

**Intra-Species PPI Network Construction.** In this task, we focus on the Human species, following the same intra-species subject used in previous studies [17, 19]. The full Human PPI graph is split into training and test sets with an 8:2 ratio, using an established protocol [46] to prevent potential data leakage. This obtains a training subgraph of 8,072 proteins and a test subgraph of 2,018 proteins. To investigate the influence of subgraph topology and size on model performance, we sample 500 subgraphs from the test set, ranging in size from 20 to 200 nodes, using each of three traversal algorithms, breadth-first search (BFS), depth-first search (DFS), and random walk (RW). Meanwhile, since existing PPI prediction models require PPI pairs for training, we sample positive and negative protein pairs from the training subgraph in a 1:1 ratio and further partition them into 80% for training

and 20% for validation, following the previous works [46]. In total, this yields 85,824 training pairs, 21,456 validation pairs, and 500 test graphs per traversal strategy for topology recovery evaluation.

**Cross-Species PPI Network Construction.** For cross-species evaluations, we use models trained on the Human PPIs to reconstruct PPI networks in the other three species: Arath, Ecoli, and Yeast. As in the intra-species setting, we sample 500 subgraphs ranging from 20 to 200 nodes from each species using BFS, DFS, and RW, resulting in 500 test networks per traversal method per species.

Subsequently, the above two tasks are evaluated using five topology-aware metrics, which are widely used in graph construction tasks (Detailed mathematical definitions are provided in Apx. D.1.4.):

- **Graph Similarity (GS) [87]** primarily quantifies the edge differences between the predicted and ground-truth graphs, as the node set in the PPI graph remains unchanged.
- **Relative Density (RD) [88]** evaluates the extent of over- or under-prediction by comparing the edge density of the predicted network with that of the ground-truth network.
- **Degree Distribution (Deg.) [89]** computes the discrepancy between node degree distributions of the predicted and ground-truth networks using maximum mean discrepancy (MMD) [90], providing a quantitative assessment of global structural differences in terms of connectivity patterns.
- **Clustering Coefficient (Clus.) [89]** uses MMD to measure the discrepancy between the distributions of local clustering coefficients in the predicted and ground-truth networks, thereby assessing the preservation of community structure.
- **Spectral [91]** calculates the discrepancy between eigenvalue spectra of normalized Laplacian matrices of predicted and true networks using MMD, reflecting global structural alignment.

### 3.2.2 Function-Oriented Task

Besides topology-based evaluation, we introduce three function-oriented tasks that are closely aligned with real-world biological applications. These tasks assess how well the reconstructed PPI networks preserve biologically meaningful properties and evaluate the practical applicability of existing models.

**Protein Complex Pathway Prediction.** Complex pathways refer to biological processes involving multiple proteins that interact with each other to perform coordinated cellular functions [92, 93], which typically form densely connected subgraphs within the larger PPI network. Accurately reconstructing these pathways can enhance our understanding of disease mechanisms and support the development of targeted therapies [94, 95]. In this task, the model first predicts pairwise protein interactions based on the input complex, then aggregates these predictions to construct predicted subgraph, which are subsequently evaluated against the ground truth. To achieve fair evaluation, we collect 235 complex pathways from Reactome [38] that share no protein overlap with the Human training graph, with pathway sizes ranging from 4 to 20 proteins. We evaluate model performance using the following metrics (Detailed descriptions are provided in Apx. D.2.1.):

- **Pathway Precision (PP)** is the proportion of predicted interactions within the complex pathway that are present in the ground truth.
- **Pathway Recall (PR)** computes the proportion of ground-truth interactions within the complex pathway that are successfully predicted.
- **Pathway Connectivity (PC)** calculates the fraction of predicted protein complex pathway subgraphs that form a connected component, reflecting biological consistency.

**GO Enrichment Analysis.** GO enrichment connects network topology with biological function, supporting gene annotation and pathway discovery [96]. This task evaluates whether reconstructed PPI networks preserve functional coherence by comparing enriched GO terms of predicted communities to those of the ground-truth network. Specifically, we reconstruct the Human test subgraph (2,018 proteins) using a trained PPI prediction model, detect communities via the Louvain algorithm [97], and perform GO enrichment using g: Profiler [98] tool across three ontologies: Molecular Function (MF), Biological Process (BP), and Cellular Component (CC). Evaluations are conducted under two metrics (Thorough definitions are included in Apx. D.2.2.):

- **Functional Alignment (FA)** depicts the average Jaccard similarity of enriched GO terms between each predicted cluster and its best-matching ground-truth partner, measuring functional alignment.
- **Consistency Ratio (CR)** is the ratio of within-cluster GO term Jaccard similarity in the predicted network to that in the ground-truth network, reflecting how well functional coherence is preserved.

**Essential Protein Justification.** Essential proteins are critical for an organism's survival and often occupy central positions in PPI networks [99, 100]. A reliable model should differentiate essential from non-essential proteins based on node degree or centrality in the reconstructed network [101]. In this task, we use the Human test subgraph (2,018 proteins) as the evaluation set and annotate essential and non-essential proteins using CCDS [102] and UniProt [37]. In detail, we select 100 essential and 100 non-essential proteins whose network centrality scores [99] differ significantly ($p < e^{-4}$). Trained models are used to reconstruct the Human test graph, and performance is assessed using the following metrics (Further details are presented in Apx. D.2.3):

- **Precision@K (P@K)** calculates the proportion of essential proteins among the top $K$ nodes ranked by network centrality scores. $K$ is set to 100 in this task.

- **Distribution Overlap (DO)** evaluates the distribution overlap between the centrality scores of essential and non-essential proteins in the reconstructed network.

# 4 Experiment

We conduct extensive experiments on five network-level tasks outlined in Sec. 3.2. Our experiments aim to answer the following research questions: **Q1:** How effectively do current models reconstruct PPI networks? (Sec. 4.3.1) **Q2**: How well do the PPI prediction models generalize? (Sec. 4.3.2) **Q3**: How do these models perform in biology practice? (Sec. 4.3.3) **Q4**: How well do standard classification metrics reflect network-level performance? (Sec. 4.3.4) **Q5**: How does the positive–negative ratio affect model performance and network reconstruction? (Sec. 4.3.5)

## 4.1 Evaluation Baseline

We consider four categories of PPI prediction models based on their design principles in our benchmark: sequence similarity-based models, naive sequence-based models, PLM-based models, and structure-based models. More details are provided in Apx. E.

**Sequence Similarity-based Model.** These models assume that protein pairs resembling known interacting pairs are more likely to interact [46]. We adopt SPRINT [52] as a representative non-deep learning method and investigate how effectively a functional PPI network can be predicted using only sequence similarity information.

**Naive Sequence-based Model.** These models implement conventional deep learning architectures [103, 104], using protein sequences as input and extracting features based on physicochemical properties or structure embeddings [46]. We evaluate three models: PIPR [48], which utilizes a residual Siamese RCNN to capture hierarchical sequence patterns; D-SCRIPT [17], which augments sequence features with structure-aware embeddings to infer inter-protein contact maps; and Topsy-Turvy [56], which integrates multi-scale sequence representations to refine interaction prediction.

**PLM-based Model.** Protein language models (PLMs) are typically trained on large protein datasets (*e.g.*, UniRef90 [105]) using self-supervised learning to produce context-aware embeddings. We consider three PLM-based models: PPITrans [19], which leverages ProtT5 [62] with multi-layer transformer blocks; TUnA [64], which applies ESM-2 [25] with an uncertainty-aware module; and PLM-interact (35M & 650M) [63], which fine-tunes ESM-2 on the STRING Human database.

**Structure-based Model.** These models either take known protein structures as input or use end-to-end structure prediction for PPI modeling. We include three structure-based methods: Struct2Graph [65] and TAGPPI [66], which use GNNs to model protein structure input, and RF2-Lite [70], a lightweight variant of RoseTTAFold2 [106] designed specifically for PPI prediction. In addition, a case study of Chai-1 [107], the newest structure prediction model, is provided in Apx. G.5.

## 4.2 Experimental Setup

We adopt the model hyperparameters as recommended in the original papers, which have shown strong performance on their respective benchmarks. We assume these configurations to be robust and effective across diverse scenarios. Due to computational limitations, PLM-interact directly uses the released pre-trained weights for inference, and RF2-Lite is evaluated on only 10% of the test graphs, while all other models are trained from scratch. The model implementation details refer to Apx. F.

Table 2: Overall results of intra-species (Human) PPI network construction task. We use three color scales of blue to denote the **first**, **second**, and **third** best performance. **RD** closer to 1 is better.

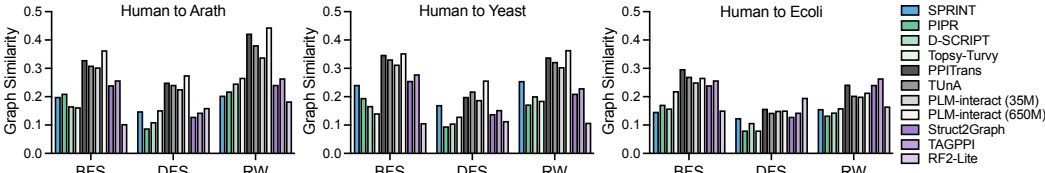

| Sampling Method | | BFS | | | | | DFS | | | | | RW | | | | | Avg. Rank |
|---|---|---|---|---|---|---|---|---|---|---|---|---|---|---|---|---|---|
| Category | Model | GS↑ | RD→ | Deg.↓ | Clus.↓ | Spectral↓ | GS↑ | RD→ | Deg.↓ | Clus.↓ | Spectral↓ | GS↑ | RD→ | Deg.↓ | Clus.↓ | Spectral↓ | |
| Seq. Sim. | SPRINT | 0.227 | 1.61 | 10.9 | 12.5 | 11.2 | 0.145 | 5.91 | 316 | 207 | 53.8 | 0.178 | 1.32 | 17.9 | 26 | 15.1 | 3 |
| Naive Sequence | PIPR | 0.209 | 4.39 | 81.7 | 40.5 | 36.2 | 0.0749 | 12.3 | 518 | 335 | 77.0 | 0.196 | 5.75 | 165 | 107 | 41.1 | 11 |
| | D-SCRIPT | 0.215 | 1.13 | 15.4 | 18.8 | 29.7 | 0.123 | 6.40 | 302 | 237 | 67.0 | 0.209 | 5.82 | 134 | 64.8 | 52 | 8 |
| | Topsy-Turvy | 0.183 | 1.74 | 17.3 | 11.4 | 14.6 | 0.130 | 3.22 | 180 | 201 | 65.9 | 0.232 | 4.88 | 104 | 64.8 | 36.7 | 5 |
| PLM | PPITrans | 0.362 | 3.39 | 52.2 | 29.1 | 26.3 | 0.314 | 4.54 | 418 | 270 | 46.6 | 0.449 | 2.57 | 71.9 | 46.1 | 19.6 | 5 |
| | TUnA | 0.342 | 2.99 | 47.8 | 30.2 | 22.9 | 0.289 | 4.44 | 416 | 272 | 46.0 | 0.450 | 2.23 | 55.4 | 42.5 | 13.7 | 4 |
| | PLM-interact (35M) | 0.383 | 2.29 | 24.4 | 16.0 | 12.8 | 0.322 | 2.85 | 209 | 78.2 | 36.3 | 0.429 | 2.47 | 52.6 | 22.6 | 13.6 | 2 |
| | PLM-interact (650M) | 0.396 | 1.64 | 30.9 | 14.8 | 16.5 | 0.350 | 2.03 | 236 | 63.6 | 42.2 | 0.491 | 1.76 | 50.3 | 21.2 | 15.4 | 1 |
| Structure | Struct2Graph | 0.27 | 5.58 | 95.1 | 50.5 | 39.2 | 0.132 | 10.8 | 508 | 321 | 67.6 | 0.237 | 5.83 | 155 | 103 | 38.3 | 10 |
| | TAGPPI | 0.283 | 4.56 | 70.8 | 36.2 | 29.5 | 0.143 | 9.94 | 491 | 309 | 65.3 | 0.256 | 4.87 | 127 | 74.2 | 28.6 | 9 |
| | RF2-Lite | 0.227 | 0.524 | 43.8 | 31.7 | 34.9 | 0.243 | 1.10 | 226 | 39.3 | 61.8 | 0.171 | 0.514 | 81.7 | 73.9 | 60.9 | 7 |

Figure 3: Cross-species generalization performance evaluated via graph similarity score.

## 4.3 Results and Analysis

### 4.3.1 From Pairs to Networks: Evaluating Structural Reconstruction

The intra-species PPI reconstruction results are summarized in Tab. 2. We can observe that: (1) PLM-based models consistently outperform other types of models; among the top five models, four are PLM-based, with PLM-interact 35M and 650M ranking first and second, respectively, highlighting that PLM can provide rich representations while capturing biologically accurate interactions; (2) most naive sequence-based and structure-based models perform poorly; for example, PIPR ranks last among the 11 baselines, while Struct2Graph ranks second to last, indicating their limited capacity to capture interaction patterns with simple architectures and features; (3) SPRINT, a non-deep learning method, ranks third, yielding a more realistic topological structure than some deep learning models, suggesting that the basic sequence similarity can still effectively preserve basic network structure; and (4) all baseline models exhibit higher RD values under DFS sampling than under BFS or RW; this is likely due to their tendency to overpredict interactions inconsistent with the underlying network topology (RD > 1), which aligns with the distribution of DFS-sampled graphs that are sparse and pathway-like, in contrast to the dense local connectivity preserved by BFS and RW.

Additionally, we find that the performance degrades as the PPI network size increases, demonstrating that existing computational methods still face challenges in reconstructing large-scale PPI networks (see Apx. G.1.4 for more details). Overall, the relative performance ranking of different model types remains consistent across traversal strategies.

**Findings.** Current PPI models tend to over-predict interactions, resulting in potential false positives. Moreover, the reconstructed PPI networks preserve low topology consistency with the ground truth: the highest graph similarity score remains below 0.5, while the other four structural metrics even deviate from their ideal minimum value by an order of magnitude. These findings highlight the need for more capable models that preserve the global structure of PPI networks beyond pairwise accuracy.

### 4.3.2 Beyond Boundaries: Generalization Across Species

The second topology-oriented task evaluates the cross-species generalization ability of PPI prediction models. As shown in Fig. 3, the overall performance decreases proportionately as the evolutionary distance from Human increases, following the order of Arath, Yeast, and Ecoli, with the average performance drops by **15.2%**, **25.3%**, and **35.2%**, respectively. In terms of model categories, PLM-based models continue to achieve the best performance among all baselines, with an average graph similarity score of 0.24 across the three species. This result indicates the robustness and generalizability of PLM-derived representations across diverse biological scenarios. In contrast, naive sequence-based models perform the worst, with an average score of only 0.17, demonstrating that simple deep learning architectures such as CNNs and RNNs are insufficient for capturing the complex evolutionary and structural signals required for cross-species.

Table 3: Overall results of function-oriented tasks. We use three color scales of blue to denote the **first**, **second**, and **third** best performance.

| Function-oriented Tasks | | Protein Complex Pathway | | | GO Enrichment | | Essentail Protein | | Avg. Rank |
|---|---|---|---|---|---|---|---|---|---|
| Category | Model | PR↑ | PP↑ | PC↑ | FA↑ | CR↑ | P@100↑ | DO↓ | |
| Seq. Sim. | SPRINT | 0.526 | 0.856 | 0.752 | 0.174 | 0.655 | 0.54 | 0.727 | 5 |
| Naive Sequence | PIPR | 0.160 | 0.588 | 0.323 | 0.182 | 0.654 | 0.52 | 0.834 | 8 |
| | D-SCRIPT | 0.225 | 0.466 | 0.316 | 0.175 | 0.678 | 0.47 | 0.629 | 6 |
| | Topsy-Turvy | 0.240 | 0.613 | 0.338 | 0.232 | 0.635 | 0.43 | 0.755 | 8 |
| PLM | PPITrans | 0.583 | 0.863 | 0.818 | 0.250 | 0.756 | 0.48 | 0.886 | 3 |
| | TUnA | 0.526 | 0.864 | 0.794 | 0.259 | 0.622 | 0.57 | 0.849 | 4 |
| | PLM-interact (35M) | 0.550 | 0.847 | 0.880 | 0.335 | 0.785 | 0.80 | 0.423 | 2 |
| | PLM-interact (650M) | 0.549 | 0.862 | 0.887 | 0.368 | 0.844 | 0.77 | 0.440 | 1 |
| Structure | Struct2Graph | 0.413 | 0.742 | 0.608 | 0.154 | 0.649 | 0.45 | 0.841 | 10 |
| | TAGPPI | 0.487 | 0.781 | 0.713 | 0.168 | 0.649 | 0.43 | 0.828 | 6 |
| | RF2-Lite | 0.554 | 0.874 | 0.135 | / | / | / | / | / |

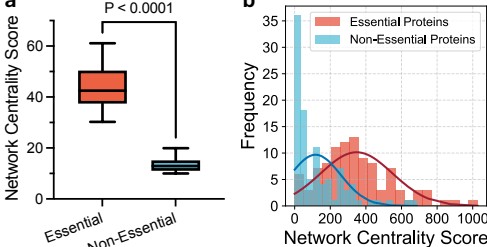

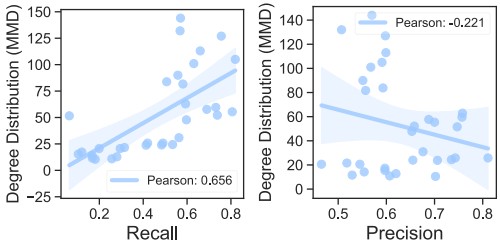

Figure 4: Essential protein analysis. (**a**) The network centrality score of essential and non-essential proteins in the ground-truth PPI network. (**b**) Network centrality score distribution of PLM-interact (650M).

Figure 5: Relationship between classification performance and graph-level metrics. The recall rate is positively correlated with the degree distribution (MMD), while the precision is negative.

**Findings.** The cross-species results highlight the challenge of transferring knowledge to evolutionarily distant species on the genetic tree [108]. This underscores the need for developing more advanced models that can capture conserved features to enable reliable transfer across species.

### 4.3.3 From Prediction to Practice: Biological Utility of PPI Models

The results of the function-oriented tasks are summarized in Tab. 3. Specifically, in the protein complex pathway reconstruction task, models tend to achieve high precision but low recall (*e.g.*, PPITrans gets a precision of 0.863 and a recall of 0.583), suggesting that predictions are accurate yet incomplete, which in turn limits the comprehensive recovery of protein complex pathways. For the GO enrichment analysis task, nearly all models perform poorly in the function alignment metric FA, with scores below 0.4, demonstrating a weak ability to preserve functional modularity in the reconstructed networks. Additionally, while most models are capable of identifying true positive hub proteins in the essential protein justification task (e.g., the PLM-interact (35M) achieves a P@100 of 0.80), they still struggle to effectively distinguish essential from non-essential proteins based on the reconstructed networks (*e.g.*, even the best-performing model, PLM-interact (650M), only attains a DO score of 0.440). As illustrated in Fig. 4, the network centrality score distributions of essential and non-essential proteins in the ground-truth graphs are well-separated. In contrast, the distributions generated by the PLM-interact (650M) exhibit substantial overlap, indicating a loss of discriminative structural features. These observations suggest that current models fail to preserve the distinguishing topological characteristics between essential and non-essential proteins in the reconstructed networks.

**Findings.** Current methods still fall short in capturing the underlying functional organization of PPI networks, leading to low functional alignment, fragmented complex pathways, and disturbed essential and non-essential proteins. These shortcomings hinder key downstream applications such as function annotation, biological module detection, and disease mechanism discovery.

### 4.3.4 Metric Alignment: Do Classification Scores Tell the Whole Story?

While standard classification metrics are commonly used to evaluate PPI models, their effectiveness in reflecting graph-level structural properties remains unclear. We investigate the correlation between

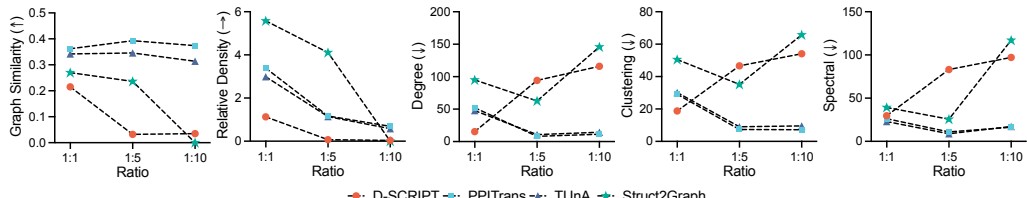

Figure 6: Ablation on the ratio of positive to negative PPI pairs. Results are shown for the BFS-based sampling configuration. For relative density, a value closer to 1 denotes better topological consistency.

traditional binary classification metrics and graph-level structural metrics, as shown in Fig. 5. Our analysis reveals an inverse relationship between recall and precision in terms of their impact on the degree distribution: higher recall increases distribution mismatch, whereas higher precision reduces it. This indicates that recall-optimized models recover more true positive interactions at the expense of false positives that distort the topology, whereas precision-focused models better preserve network structure. Additional correlation analyses are provided in Apx. G.1.3.

**Findings.** Standard classification metrics cannot completely reflect network-level structure, revealing the necessity for graph-aware evaluations in robust PPI modeling.

### 4.3.5 Data Matters: Do Positive to Negative Ratios Affect Network Construction?

Considering that real PPI networks are inherently sparse, we vary the positive-to-negative ratio of training pairs (1:1 to 1:10) under the BFS-based sampling configuration to assess the effect of class imbalance (Fig. 6). Since SPRINT does not require negative samples for training and the PLM-interact models are fixed for inference, we include four representative methods in this analysis. The complete results across three sampling strategies are provided in Apx. G.3.

For D-SCRIPT and Struct2Graph, increasing the ratio severely degrades performance—the models tend to underpredict interactions, with RD scores approaching 0. In contrast, PPITrans and TUnA show improved results when the ratio increases to 1:5, suggesting moderate imbalance regularizes learning, but their performance drops again at 1:10 as excessive negatives bias the networks toward overly sparse topologies.

**Findings.** Moderate class imbalance can enhance the performance of stronger models such as PLM-based architectures by providing richer negative supervision and better reflecting the sparsity of real PPI networks. However, the optimal positive-to-negative ratio remains task- and architecture-dependent, warranting further investigation into more advanced strategies for biologically realistic network learning.

## 5 Conclusion

In this paper, we present PRING, the first comprehensive benchmark designed to evaluate PPI prediction models beyond pairwise classification, focusing on their ability to reconstruct biology-aware PPI networks. By introducing topology and function-oriented tasks across multiple species with well-designed strategies on both data redundancy and leakage, PRING enables a more rigorous and application-driven assessment of model performance. Extensive experiments reveal that current models typically fail to preserve the global structure and biological coherence of interaction networks. These findings emphasize the limitations of traditional evaluation protocols and highlight the need for more holistic and biology-informed approaches. We believe PRING is a valuable resource for developing more reliable PPI prediction models that support real-world biological discovery.

## Acknowledgment

This work was partially supported by the New Generation Artificial Intelligence-National Science and Technology Major Project of China (2025ZD0121801). This work was also partially supported by the Ministry of Education, Singapore (MOE T1251RES2309, MOE T2EP20125-0039) and the Agency for Science, Technology and Research (A*STAR H25J6a0034). This work was also partially supported by the JC STEM Lab of AI for Science and Engineering, funded by The Hong Kong Jockey Club Charities Trust, the Research Grants Council of Hong Kong (Project No. CUHK14213224). This work was also partially supported by Shanghai Artificial Intelligence Laboratory.

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

# Appendix

# A  Data and Code Availability

## A.1  `PRING` Dataset

The `PRING` dataset is available at `https://huggingface.co/datasets/piaolaidangqu/PRING`.

## A.2  Evaluation Code

The code for benchmarking PPI prediction models on `PRING` is available at `https://github.com/SophieSarceau/PRING`.

## A.3  Project Website

The `PRING` project website is available at `https://zhanglab.comp.nus.edu.sg/PRING/`.

# B  Limitations & Future Work

## B.1  Data Scope

Though `PRING` currently benchmarks PPI prediction on four model organisms, including Human, Ecoli, Yeast, and Arath, which are widely studied, it does not cover the full phylogenetic diversity of life. As a result, the generalizability of PPI prediction models to non-model or underrepresented taxa remains unexplored. To enable a more comprehensive evaluation, future extensions of `PRING` will focus on expanding its taxonomic coverage by curating high-quality PPIs data from additional clades (*e.g.*, archaea, non-model metazoans), enabling a more comprehensive evaluation of evolutionary transfer and model robustness across the tree of life. Moreover, certain biological applications, such as antiviral drug development, require the modeling of PPIs between Human and pathogen proteins. To support such use cases, future versions of `PRING` will incorporate Human–virus PPI networks, advancing the benchmark's utility in biomedical research.

## B.2  Protein–Protein Interaction Scope

While `PRING` develops holistic downstream tasks to evaluate the model's capability, all tasks currently treat PPIs as binary edges, ignoring other biological details such as interaction types (*e.g.*, activation, inhibition). This would limit the ability to capture regulatory and signaling mechanisms. Moreover, PPIs are often dynamic and context-specific, varying across tissues, cell types, and conditions. Modeling them as static interactions overlooks this critical aspect of biological systems. For future work, we plan to extend `PRING` to support multi-type interactions and incorporate conditional PPIs under different biological contexts. This will enable more accurate and versatile benchmarking for real-world applications.

# C  Dataset Documentation

## C.1  Data Sources

In this work, we construct the PPI benchmark dataset using data integrated from multiple public data sources. For `PRING` PPI network, we select UniProt, Reactome, IntAct, and STRING as our primary source databases. These databases are chosen based on the following considerations: 1) They collectively provide high-quality annotations and broad biological coverage; 2) Most other PPI databases either overlap substantially with these sources (*e.g.*, BioGrid [109], MINT [110]) or exhibit limited scope and lower data reliability (*e.g.*, HuRI [111], PrePPI [112]); 3) These four resources are widely recognized and commonly adopted within the bioinformatics community [113, 114].

For function-oriented tasks, the data includes Gene Ontology (GO), and the Consensus CDS (CCDS) project. License information for each source is provided in Tab. 4.

**UniProt.**  UniProt is a well-known comprehensive resource for protein sequence and functional information. In our work, we fetch PPIs and species information through manually reviewed entries from Swiss-Prot – a section of UniProt to ensure high-quality protein annotations. The raw data are available at:

- `https://ftp.uniprot.org/pub/databases/uniprot/current_release/knowledgebase/complete/uniprot_sprot.dat.gz`
- `https://ftp.uniprot.org/pub/databases/uniprot/current_release/knowledgebase/complete/uniprot_sprot.fasta.gz`

**Reactome.** Reactome is a curated knowledgebase of biological pathways and reactions. We take their high-quality PPIs to construct our dataset, and additionally collect protein - complex and pathway information to support the **protein complex pathway prediction** task (Sec. D.2.1). The raw data are available at:

- `https://reactome.org/download/current/interactors/reactome.all_species.interactions.tab-delimited.txt`
- `https://reactome.org/download/current/ComplexParticipantsPubMedIdentifiers_human.txt`
- `https://reactome.org/download/current/Complex_2_Pathway_human.txt`
- `https://reactome.org/download/current/UniProt2Reactome.txt`

**IntAct.** IntAct provides a free, open-source database system and analysis tools for molecular interaction data. We collect PPIs from IntAct to construct our dataset. The raw data of IntAct is available at:

- `https://ftp.ebi.ac.uk/pub/databases/intact/current/psimitab/intact.zip`

**STRING.** STRING is a database of known and predicted PPIs, integrating evidence from multiple sources. Following common practice, we include only high-confidence interactions with their Combine Score > 0.7 in our dataset. The raw data is available at:

- `https://stringdb-downloads.org/download`

**Gene Ontology.** The Gene Ontology (GO) knowledgebase is the world's largest source of information on the functions of genes, which provides a structured vocabulary for annotating gene and protein functions across species. We extract GO terms related to our PPI networks to support the **GO enrichment analysis** task (Sec. D.2.2). The raw data is available at:

- `http://current.geneontology.org/annotations/filtered_goa_uniprot_all_noiea.gpad.gz`

**CCDS.** The Consensus CDS (CCDS) project provides a curated set of protein-coding regions that are consistently annotated across major genome databases. We fetch reliably annotated proteins from CCDS for our **essential protein justification** task (Sec. G.2.3). The raw data is available at:

- `https://ftp.ncbi.nlm.nih.gov/pub/CCDS/current_human/`

Table 4: Licenses of datasets used in this work.

| Dataset | Usage | Licence | URL |
|---|---|---|---|
| **UniProt** | PPI dataset construction (curated data) | CC BY 4.0 | `https://www.uniprot.org/help/license` |
| **Reactome** | PPI dataset construction (curated data) and relation information for **Protein Complex Pathway Prediction** task | CC0 1.0 Universal | `https://reactome.org/about/news` |
| **IntAct** | PPI dataset construction (curated data) | CC BY 4.0 | `https://www.ebi.ac.uk/intact/download` |
| **STRING** | PPI dataset construction (high confidence data) | CC BY 4.0 | `https://string-db.org/cgi/access?footer_active_subpage=licensing` |
| **Gene Ontology** | Functional annotation for **GO Enrichment Analysis** task | CC BY 4.0 | `https://geneontology.org/docs/go-citation-policy` |
| **CCDS** | Protein information for **Essential Protein Justification** task | License not specified (redistributed via Bioregistry under CC BY 4.0) | `http://www.ncbi.nlm.nih.gov/CCDS` |

## C.2 Collection Process

Our data collection process starts from UniProt, we first download the complete Swiss-Prot entries dat file and parse each protein record to identify the "CC -!- INTERACTION" section, e.g:

```
CC   -!- INTERACTION:
CC       P12345; Q8N1H7:XYZ_HUMAN (xeno); Xeno interaction [EMBL:ABC123];
CC       P12345; Q9Y6K1:DEF_HUMAN; NbExp=3; IntAct=EBI-12345, EBI-67890;
```

If the interacting partners is a Swiss-Prot protein, we extract the interaction as a PPI entry. Additionally, we download the full Swiss-Prot FASTA file containing 573,230 protein sequences. Using the NCBI taxonomy ID specified in the FASTA headers (e.g., OX=9606), we extract protein sets for four representative species: Human (Homo sapiens, 9606), Arath (Arabidopsis thaliana, 3702), Yeast (Saccharomyces cerevisiae, 559292), and Ecoli (Escherichia coli, 83333).

We then download the complete contents file from IntAct and the Protein–Protein Interaction file from Reactome, extract all PPI entries, and filter them based on our constructed protein sets to retain only Swiss-Prot proteins from the four selected species. PPIs derived from UniProt, IntAct, and Reactome are considered curated, high-quality data. To further expand our dataset, we also download species-specific subsets from STRING for the four selected species, and retain only interactions with a combined score greater than 0.7 as high-confidence PPIs. These are merged with the curated PPIs to form our complete raw dataset.

To ensure fair evaluation of PPI models and minimize data redundancy, we first filter proteins to retain only those with sequence lengths between 50 and 1000. We then apply MMSeqs2 to cluster protein sequences and keep only those pairs with sequence identity $\leq 40\%$. Additionally, we remove proteins that share the same entry name across different species to eliminate functionally redundant cross-species homologs.

## C.3 Statistical Analysis

To ensure transparency in dataset construction, we first report the detailed preprocessing steps for building the PPI benchmark. After retrieving interaction data from four curated databases, we first apply a protein-level filter (Fig. 2) to retain only experimentally verified proteins with annotated functions (SwissProt) and belonging to one of the four target species. This filtering step results in 15,043 Human, 13,271 Arath, 5,232 Yeast, and 3,855 E. coli proteins.

We then perform sequence similarity filtering ($\leq 40\%$) using MMSeqs2, followed by function-based redundancy removal to eliminate homologous or functionally overlapping proteins across species.

- **Arath**: 7,487 proteins were removed by MMSeqs2, and 759 were excluded due to shared functions with Human, yielding 5,025 final proteins ($\downarrow 62\%$).

- **Yeast**: 1,033 proteins were filtered by MMSeqs2, and 1,026 shared functions with Human, leaving 3,173 proteins ($\downarrow 39\%$).

- **Ecoli**: 452 proteins were removed by MMSeqs2, and 207 due to function overlap, resulting in 3,196 proteins ($\downarrow 17\%$).

After these filtering steps, we obtain the final `PRING` PPI network dataset.

Then, we conduct a statistical analysis of the fully processed PPI datasets obtained through the above pipeline, with results summarized as follows:

Fig. 7 illustrates the distributions of protein sequence lengths and node degrees across the four species in our processed PPI dataset. Protein lengths are constrained between 50 and 1000 amino acids, with most sequences falling in the 250–500 aa range, resembling a roughly normal distribution. The Human PPI network, benefiting from richer data coverage, exhibits a smoother and broader degree distribution with generally higher values. In contrast, the other three species show lower node degrees overall, indicating sparser interaction networks.

Tab. 5 summarizes key statistics of the processed PPI datasets across the four selected species. The number of proteins and interactions varies notably after the filter process, with Human (contributing the training set) having the largest network (129,861 interactions among 10,090 proteins), while Yeast and Ecoli have more compact networks. In terms of the topology, Human exhibits the smallest diameter and average path length, suggesting a denser and more interconnected PPI network. In

Table 5: Summary statistics of processed PPI dataset across species.

| Species | PPI Count | Protein Count | Avg. Degree | Avg. Seq. Len | Diameter | Avg. Path Len |
|---|---|---|---|---|---|---|
| **Human** | 129,861 | 10,090 | 25.74 | 429.55 | 11 | 3.64 |
| **Arath** | 23,584 | 5,025 | 9.39 | 417.10 | 15 | 5.62 |
| **Yeast** | 15,921 | 3,173 | 10.04 | 413.52 | 19 | 5.23 |
| **Ecoli** | 17,452 | 3,196 | 10.92 | 310.67 | 14 | 4.81 |

contrast, Arath shows a larger diameter and average shortest path length, indicating a more fragmented or modular interaction structure.

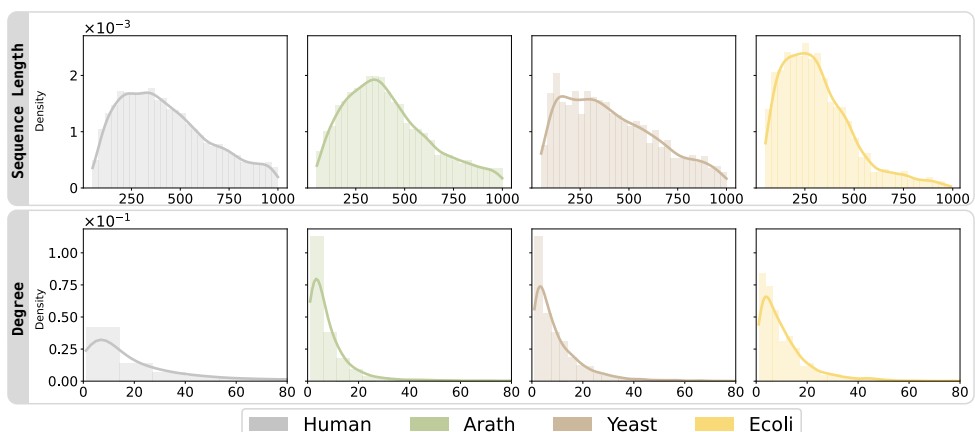

Figure 7: Distribution of protein sequence lengths (top row) and node degrees in the PPI network (bottom row) across four species in our dataset. (The y-axis indicates the density for both distributions)

## D    Task Documentation

This section outlines the formal definitions of each task included in `PRING` and details the corresponding evaluation metrics used to assess model performance.

### D.1    Topology-oriented Task

This task assesses whether PPI prediction models can learn and preserve the underlying topological structure of PPI networks. To this end, `PRING` requires each model to reconstruct the PPI network from pairwise predictions and evaluates the result using graph-level metrics.

Formally, given a PPI graph with $N$ proteins, each protein is represented by features $x_i = \{s_i, c_i\}$ for $i = 1, \ldots, N$, where $s_i$ denotes the amino acid sequence and $c_i$ denotes any auxiliary context information. A model $f$ predicts an interaction label for each protein pair $(i, j)$, where $1 \leq i < j \leq N$, as follows:

$$I_{i,j}^{\text{pred}} = f(x_i, x_j) \in \{0, 1\}. \tag{2}$$

Here, $I_{i,j}^{\text{pred}} = 1$ indicates a predicted interaction, and $0$ indicates no interaction.

The reconstructed PPI network is then formed by aggregating all predicted positive interactions:

$$G^{\text{pred}} = \text{Aggregate}\big\{(i, j) \mid I_{i,j}^{\text{pred}} = 1,\ 1 \leq i < j \leq N\big\}. \tag{3}$$

Finally, the reconstructed graph $G^{\text{pred}}$ is evaluated against the ground-truth graph $G^{\text{true}}$ using a suite of graph-level metrics detailed below.

We consider two subtasks to evaluate the intra-species and cross-species capabilities of PPI prediction models. Below, we detail the task definitions and corresponding evaluation protocols.

### D.1.1    Comparison of Different Traversal Strategies

Since our primary focus is on protein–protein interaction (PPI) networks, we consider three widely used graph traversal strategies for sampling: breadth-first search (BFS), depth-first search (DFS), and

random walk (RW). Each of these strategies captures different topological and biological characteristics of PPI networks. The distinct features they emphasize are summarized below, as shown in Tab. 6.

Table 6: PPI subgraph structural characteristics and biological network types corresponding to different sampling strategies.

| Sampling Strategy | Biological Network Structure | Functional Module Example | Citation |
|---|---|---|---|
| BFS | Locally dense structures, such as protein complexes, modules | Ribosomal protein complex, intra-cellular signaling module | [115] |
| DFS | Chain/tree structures, such as signaling or metabolic pathways | MAPK signaling pathway, glycoly-sis metabolic pathway | [116] |
| RW | Hub-dominant structures, such as regulatory networks and pan-functional networks | TP53 regulatory network, transcription factor network | [117] |

### D.1.2 Intra-species PPI network Construction

This task focuses on the Human species, following the same intra-species setting used in prior studies [17, 19]. The full Human PPI graph is partitioned into training and test subgraphs with an 8:2 split, using a leakage-free protocol to avoid data contamination. This results in a training subgraph of 8,072 proteins and a test subgraph of 2,018 proteins with no protein overlap. To examine how subgraph topology and size affect model performance, we sample 500 subgraphs from the test sets, each ranging from 20 to 200 nodes, using three traversal algorithms: breadth-first search (BFS), depth-first search (DFS), and random walk (RW). These traversal strategies can be associated with distinct biological network types. BFS captures densely connected local neighborhoods, making it suitable for modeling protein complexes and functional modules characterized by high intra-cluster interaction density [115]. DFS, in contrast, tends to generate chain-like or tree-structured subgraphs, resembling the topology of signaling or metabolic pathways, where interactions follow a directional flow [116]. Random walk explores the network probabilistically and is more likely to visit hub proteins, aligning with regulatory or hub-based networks that reflect transcriptional or cellular control systems [117, 118]. By leveraging these biologically motivated sampling strategies, we systematically evaluate how PPI models generalize across different local topologies.

Since existing PPI models are trained on protein pairs, we generate training data by sampling positive and negative protein pairs from the training subgraph in a 1:1 ratio, and further divide them into 80% for training and 20% for validation, following protocols in [46]. In total, this yields 85,824 training pairs, 21,456 validation pairs, and 500 test subgraphs per traversal strategy for topological evaluation.

### D.1.3 Cross-species PPI network Construction

To assess cross-species generalization, models trained on Human PPIs are used to reconstruct networks in three additional species: *Arath*, *Ecoli*, and *Yeast*. For each species, we sample 500 subgraphs ranging from 20 to 200 nodes using BFS, DFS, and RW, yielding 500 test graphs per traversal method per species for evaluation.

### D.1.4 Evaluation Metrics

Both intra-species and cross-species PPI network construction tasks are evaluated under five graph-level metrics, which are widely used in graph-generation tasks.

**Graph Similarity (GS) [87].** Graph similarity primarily quantifies the edge differences between the predicted and ground-truth graphs, as the node set in the PPI graph remains unchanged. Mathematically, it is defined as:

$$\text{Graph Similarity} = 1 - \frac{\|\hat{A} - A\|_1}{|E| + |\hat{E}|}, \tag{4}$$

where $A \in \{0,1\}^{N \times N}$ is the adjacency matrix of the ground-truth graph $G^{\text{true}}$, $\hat{A} \in \{0,1\}^{N \times N}$ is the adjacency matrix of the predicted graph $G^{\text{pred}}$. $E$ and $\hat{E}$ are the sets of edges in the ground-truth and predicted graphs, respectively.

This metric ranges between 0 and 1, and a higher value indicates better alignment between the prediction and the ground truth.

**Relative Density (RD) [88].** Relative density evaluates the extent of over- or under-prediction by comparing the edge density of the predicted network with that of the ground-truth network.

$$\text{Relative Density} = \frac{|\hat{E}|/\binom{N}{2}}{|E|/\binom{N}{2}} = \frac{|\hat{E}|}{|E|} \tag{5}$$

where $N$ is the number of proteins in the PPI network, and $|E|$ and $|\hat{E}|$ denote the number of edges in the ground-truth and predicted graphs, respectively.

A value of RD $> 1$ indicates over-prediction, while RD $< 1$ suggests under-prediction. An RD of 1 implies that the predicted graph has the same edge density as the ground truth.

**Degree Distribution (Deg.) [89].** Degree distribution computes the discrepancy between node degree distributions of the predicted and ground-truth networks using maximum mean discrepancy (MMD) [90], providing a quantitative assessment of global structural differences in terms of connectivity patterns. However, the direct output of MMD lacks a reference scale and does not reflect the relative extent of discrepancy. To address this limitation, we follow prior work [89] and report a normalized ratio:

$$\text{Degree Distribution (MMD)} = \frac{\text{MMD}^2(\text{pred}, \text{test})}{\text{MMD}^2(\text{test}, \text{test})}, \tag{6}$$

where **pred** and **test** denote sets of predicted and ground-truth PPI networks, respectively. Each network is first transformed into a degree histogram (*i.e.*, a vector summarizing its node degree distribution), and the $\text{MMD}^2$ is then computed over these histogram sets.

Ideally, the range of Deg. is equal to or larger than 1. A lower value of Deg. indicates better alignment of degree distributions, with a value close to 1 suggesting that the predicted networks are as similar to the test set as the test networks are to themselves.

**Clustering Coefficient (Clus.) [89].** Clustering coefficient uses MMD to measure the discrepancy between the distributions of local clustering coefficients in the predicted and ground-truth networks, thereby assessing the preservation of community structure.

As with degree distribution, we compute the discrepancy in a relative form by normalizing the $\text{MMD}^2$ between the predicted and test sets. Each network is transformed into a histogram of local clustering coefficients, and the MMD is applied over these aggregated distributions.

**Spectral [91].** Spectral calculates the discrepancy between eigenvalue spectra of normalized Laplacian matrices of predicted and true networks using MMD, reflecting global structural alignment.

Again, we report the relative discrepancy by normalizing the $\text{MMD}^2$ between predicted and test sets. Each network is represented by a vector of eigenvalues of its normalized Laplacian matrix, and MMD is applied over the resulting spectral distributions.

## D.2 Function-oriented Task

Besides topology-based evaluation, we introduce three function-oriented tasks closely aligned with real-world biological applications. These tasks assess how well the reconstructed PPI networks preserve biologically meaningful properties and evaluate the practical applicability of existing models.

### D.2.1 Protein Complex Pathway Prediction

Complex pathways refer to biological processes involving multiple proteins that interact with each other to perform coordinated cellular functions, which typically form densely connected subgraphs within the larger PPI network. Accurately reconstructing these pathways can enhance our understanding of disease mechanisms and support the development of targeted therapies [94, 95]. In this task, the model first predicts pairwise protein interactions based on the input protein complex pathways, then aggregates these predictions to construct predicted subgraphs $G^{\text{pred}}$, which are subsequently evaluated against the ground truth $G^{\text{true}}$. For fair evaluation, we curate 235 human protein complex pathways from Reactome [38], ensuring no protein overlap with the Human training graph. Pathway sizes range from 4 to 20 proteins.

The model's performance is evaluated using the following metrics:

**Pathway Precision (PP).** The proportion of predicted interactions within the complex pathway that are also present in the ground-truth subgraph:

$$\text{Pathway Precision} = \frac{|E^{\text{pred}} \cap E^{\text{true}}|}{|E^{\text{pred}}|}, \tag{7}$$

where $E^{\text{pred}}$ and $E^{\text{true}}$ represent the sets of predicted and ground-truth interactions, respectively.

**Pathway Recall (PR).** The proportion of ground-truth interactions within the complex pathway that are successfully recovered by the prediction:

$$\text{Pathway Recall} = \frac{|E^{\text{pred}} \cap E^{\text{true}}|}{|E^{\text{true}}|}, \tag{8}$$

where $E^{\text{pred}}$ and $E^{\text{true}}$ represent the sets of predicted and ground-truth interactions, respectively.

**Pathway Connectivity (PC).** The fraction of predicted pathway subgraphs that form a single connected component, reflecting biological plausibility:

$$\text{Pathway Connectivity} = \frac{1}{N} \sum_{i=1}^{N} \left[ G_i^{\text{pred}} \text{ is connected} \right], \tag{9}$$

where $N$ is the total number of pathways.

### D.2.2 GO Enrichment Analysis

GO enrichment connects network topology with biological function, supporting gene annotation and pathway discovery [96]. This task evaluates whether reconstructed PPI networks preserve functional coherence by comparing enriched GO terms of predicted communities to those of the ground-truth network. Specifically, we reconstruct the Human test subgraph (2,018 proteins) using a trained PPI prediction model, detect communities via the Louvain algorithm [97], and perform GO enrichment using g: Profiler [98] tool across three ontologies: Molecular Function (MF), Biological Process (BP), and Cellular Component (CC).

Evaluations are conducted under two metrics:

**Functional Alignment (FA).** Measures the average Jaccard similarity between the enriched GO terms of each predicted cluster and its best-matching ground-truth cluster:

$$\text{Functional Alignment} = \frac{1}{K} \sum_{i=1}^{K} \max_{j} \frac{|T_i^{\text{pred}} \cap T_j^{\text{true}}|}{|T_i^{\text{pred}} \cup T_j^{\text{true}}|}, \tag{10}$$

where $K$ is the number of predicted clusters, and $T_i^{\text{pred}}$, $T_j^{\text{true}}$ are the sets of enriched GO terms in the $i$th predicted cluster and $j$th ground-truth cluster, respectively.

A higher functional alignment score ($0 \leq \text{FA} \leq 1$) indicates better functional alignment between predicted and ground-truth communities

**Consistency Ratio (CR).** Compares the average within-cluster GO term similarity in the predicted network to that in the ground-truth network:

$$\text{CR} = \frac{1}{K} \sum_{i=1}^{K} \frac{\text{Jaccard}(T_i^{\text{pred}})}{\max_{j} \text{Jaccard}(T_j^{\text{true}})}, \tag{11}$$

where $K$ is the number of predicted clusters, $T_i^{\text{pred}}$ denotes the set of GO terms associated with proteins in the $i$th predicted cluster, and $\text{Jaccard}(T_i)$ is the average pairwise Jaccard similarity between the GO term sets of all protein pairs within cluster $i$. For each predicted cluster $i$, the denominator uses the best-matching ground-truth cluster $j$.

A consistency ratio ($0 \leq \text{CR} \leq 1$) close to 1 indicates stronger preservation of functional coherence within clusters.

### D.2.3 Essential Protein Justification

Essential proteins are critical for an organism's survival and often occupy central positions in PPI networks [99, 100]. A reliable model should differentiate essential from non-essential proteins based on node degree or centrality in the reconstructed network [101]. In this task, we use the Human test subgraph (2,018 proteins) as the evaluation set and annotate essential and non-essential proteins using CCDS [102] and UniProt [37]. Specifically, we select 100 essential and 100 non-essential proteins whose centrality scores [99] differ significantly, with $p$ value less than $10^{-4}$ (See Fig. 4(a)).

Trained models are used to reconstruct the Human test graph, and performance is assessed using the following metrics:

**Precision@K (P@K).** Measures the proportion of essential proteins among the top $K$ nodes ranked by network centrality scores:

$$\text{Precision@K} = \frac{|\text{Top}_K \cap \text{Essential}|}{K} \tag{12}$$

where $\text{Top}_K$ denotes the set of top-$K$ nodes ranked by centrality in the reconstructed graph, and Essential is the set of ground-truth essential proteins. We set $K = 100$ in this study.

**Distribution Overlap (DO).** Quantifies the distribution overlap between the centrality score of essential and non-essential proteins using the area under the minimum of their probability density functions (PDFs):

$$\text{Distribution Overlap} = \int_{-\infty}^{\infty} \min\left(p_{\text{essential}}(x),\, p_{\text{non-essential}}(x)\right) dx \tag{13}$$

where $p_{\text{ess}}(x)$ and $p_{\text{non}}(x)$ are the estimated PDFs of the centrality scores for essential and non-essential proteins, respectively.

Lower values of distribution overlap indicate better separability between essential and non-essential proteins, while higher P@K reflects stronger prioritization of biologically important proteins.

## E  Baseline Models

This section introduces the baseline models evaluated in our benchmark.

### E.1  Sequence Similarity-based Method

**SPRINT [52].**  SPRINT is a high-throughput, alignment-based method that identifies PPIs by searching for local sequence similarities between query proteins and known interacting pairs. It uses a spaced seed hashing mechanism to locate short, conserved motifs and then filters out nonspecific matches to improve precision. By avoiding supervised learning, it remains model-free and does not require negative sampling, making it computationally efficient to full proteome predictions. We access the codebase via `https://github.com/lucian-ilie/SPRINT`.

### E.2  Naive Sequence-based Method

**PIPR [48].** PIPR employs a Siamese residual RCNN to learn complex interaction features directly from protein sequences. Each protein is embedded using a property-aware amino acid encoding, followed by convolutional and recurrent layers to extract both short-term motifs and long-range dependencies. PIPR is trained end-to-end to predict whether two sequences interact, making it effective for pairwise classification, though limited in capturing higher-level network structures. The source code is provided in `https://github.com/muhaochen/seq_ppi`.

**D-SCRIPT [17].** D-SCRIPT bridges the gap between sequence and structure by predicting physical interactions via inferred inter-residue contact maps. It uses pretrained structure-aware embeddings, which are then projected into a contact space using a convolutional scoring module. A key innovation is its regularization mechanism that enforces structural plausibility, even without access to actual 3D structures. The prediction of the contacts between proteins enables its interpretability. We use the recommended implementation method in `https://d-script.readthedocs.io/en/stable/usage.html`.

**Topsy-Turvy [56].** Topsy-Turvy integrates bottom-up sequence modeling with top-down global network patterns. During training, it incorporates graph-derived supervision to infuse sequence embeddings with knowledge of network-level interactions. The model retains only sequence input at test time, allowing practical use in non-model organisms. We use the recommended implementation method in `https://d-script.readthedocs.io/en/stable/usage.html`.

### E.3 Protein Language Model-based Method

**PPITrans [19].** PPITrans leverages protein language models (PLMs) for generalizable PPI prediction, using ProtT5 [62] to encode contextual sequence embeddings. A transformer encoder is applied to model pairwise interactions from these embeddings. The model demonstrates strong cross-species performance, suggesting that PLM-derived representations capture semantic and functional signals relevant to interaction propensity and are transferable across evolutionary distances. We follow the codebase in `https://github.com/LtECoD/PPITrans`.

**TUnA [64].** TUnA extends PLM-based PPI prediction by incorporating uncertainty estimation. It uses ESM-2 embeddings [25] as input to transformer layers and employs a spectral-normalized neural Gaussian process as the output layer to quantify prediction confidence. This design enables TUnA to provide calibrated interaction scores, which are particularly valuable for prioritizing high-confidence predictions in large-scale or experimental screening settings. Implementation is followed by `https://github.com/Wang-lab-UCSD/TUnA`.

**PLM-interact [63].** PLM-interact jointly encodes protein pairs by concatenating sequences and feeding them into ESM-2 [25], enabling attention mechanisms to directly model inter-sequence dependencies. This paired encoding strategy captures interaction-specific features, such as co-evolutionary signals and cross-residue contacts, allowing the model to effectively learn the "language of interactions". The official implementation is provided in `https://github.com/liudan111/PLM-interact`.

### E.4 Structure-based Method

**Struct2Graph [65].** Struct2Graph models PPIs using graph neural networks (GNNs) over residue-level graphs constructed from 3D protein structures. It employs a dual GNN encoder with shared weights and a mutual attention mechanism to highlight interfacial regions between proteins. By capturing spatial geometry and residue connectivity, the model effectively identifies structural determinants of binding. A key limitation, however, is its reliance on experimentally resolved structures from the Protein Data Bank (PDB), which constrains its applicability to proteins with unknown 3D conformations. The source code is provided in `https://github.com/baranwa2/Struct2Graph`.

**TAGPPI [66].** TAGPPI introduces structure-aware learning by combining predicted contact maps with sequence features. It uses residue–residue graphs as input to a GNN and supplements them with a 1D CNN over the raw sequence. The fusion of these modalities enables TAGPPI to benefit from structural topology even when only sequence input is available. The model outperforms prior sequence-only methods and shows robustness across different organisms, validating the utility of predicted structures for downstream functional tasks. The source code is given in `https://github.com/xzenglab/TAGPPI`.

**RF2-Lite [70].** RF2-Lite is a lightweight derivative of RoseTTAFold2 [106] tailored for high-throughput PPI screening. It uses paired multiple sequence alignments (MSAs) and a reduced multitrack architecture to predict inter-chain distance maps efficiently. Although less accurate than full-fledged structure predictors, RF2-Lite achieves strong performance with dramatically lower inference time, making it suitable for proteome-scale predictions. We follow the recommended implementation in `https://github.com/SNU-CSSB/RF2-Lite`.

**Chai-1 [107].** Chai-1 is a state-of-the-art, multi-modal foundation model developed by the Chai Discovery team for molecular structure prediction. It is highly effective at predicting the structures of proteins, small molecules, DNA, RNA, and covalently modified compounds. Unlike other structure prediction models that rely heavily on multiple sequence alignments (MSAs), Chai-1 can perform effectively in single-sequence mode while maintaining high accuracy. A standout feature of Chai-1 is its ability to incorporate experimental data, such as epitope conditioning, to enhance prediction accuracy, particularly in antibody-antigen interactions. This integration can significantly improve performance, making it a valuable tool in antibody engineering and drug discovery. We use the source code provided in `https://github.com/chaidiscovery/chai-lab`.

## F   Experimental Settings

We follow the recommended hyperparameters provided in each model's official codebase, as these settings have demonstrated strong performance on their respective benchmarks. We assume these configurations to be robust and generalizable across diverse evaluation scenarios.

Specifically, PLM-interact (35M and 650M variants) is used in inference mode with publicly released pre-trained weights. Due to computational constraints, RF2-Lite is evaluated on only 10% of the test graphs (a graph is randomly sampled for each node size for evaluation). All other baseline models are trained from scratch. In addition, we conduct a case study of Chai-1 on just three Human graphs (node sizes range from 20 to 60). For the binary-based PPI prediction models, we use the default probability threshold of 0.5 to distinguish between interacting and non-interacting pairs.

The experiments are conducted on heterogeneous hardware platforms. For the SPRINT, which does not need GPUs for inference, the experiments are done on Kunpeng-920, a 64-core platform. For the model trained from scratch, NVIDIA A100-PCIE-40GB is used with the same CPU support. Additionally, for the RF2-Lite and Chai-1, the experiments are conducted on NVIDIA-A800-SXM4-80GB. Since RF2-Lite requires MSAs as input, we use Uniref90 [105] as the search database with GPU-accelerated version of MMseq2 [119]. Given that Chai-1 performs well without the use of MSAs or templates, and that generating MSAs is computationally intensive, we opt to use single-sequence inputs exclusively.

## G   Experimental Results

We present detailed results for all evaluated models across both topology-oriented (Sec. G.1) and function-oriented tasks (Sec. G.2). The following sections offer a comprehensive comparison of model performance, highlighting their ability to preserve structural properties and capture biologically meaningful patterns.

For topology-oriented tasks, we additionally report standard classification metrics for all baseline models to provide complementary insights. The test dataset used for classification follows the evaluation protocol established in prior work [46].

We further include the complete test results from the ablation study on varying positive-to-negative PPI training ratios in Sec. G.3. We analyze the impact of different probability thresholds on the trade-off between precision and recall in Sec. G.4 as well.

Additionally, we include a case study on Chai-1 [107] in Sec. G.5, a state-of-the-art protein structure prediction model, to evaluate its potential in PPI network construction.

Finally, we conduct a scaling-law analysis in Sec. G.6 to investigate how model size correlates with graph-level performance, offering insights into the trade-offs between model complexity and predictive accuracy.

### G.1   Topology-oriented Task

Table 7: Graph-level results of intra-species (Human) PPI network construction task. We use three color scales of blue to denote the **first**, **second**, and **third** best performance. **RD** closer to 1 is better.

| Sampling Method | | BFS | | | | | DFS | | | | | RW | | | | | Avg. Rank |
|---|---|---|---|---|---|---|---|---|---|---|---|---|---|---|---|---|---|
| Category | Model | GS↑ | RD→ | Deg.↓ | Clus.↓ | Spectral↓ | GS↑ | RD→ | Deg.↓ | Clus.↓ | Spectral↓ | GS↑ | RD→ | Deg.↓ | Clus.↓ | Spectral↓ | |
| Seq. Sim. | SPRINT | 0.227 | 1.61 | 10.9 | 12.5 | 11.2 | 0.145 | 5.91 | 316 | 207 | 53.8 | 0.178 | 1.32 | 17.9 | 26 | 15.1 | 3 |
| Naive Sequence | PIPR | 0.209 | 4.39 | 81.7 | 40.5 | 36.2 | 0.0749 | 12.3 | 518 | 335 | 77.0 | 0.196 | 5.75 | 165 | 107 | 41.1 | 11 |
| | D-SCRIPT | 0.215 | 1.13 | 15.4 | 18.8 | 29.7 | 0.123 | 6.40 | 302 | 237 | 67.0 | 0.209 | 5.82 | 134 | 64.8 | 52 | 8 |
| | Topsy-Turvy | 0.183 | 1.74 | 17.3 | 11.4 | 14.6 | 0.130 | 3.22 | 180 | 201 | 65.9 | 0.232 | 4.88 | 104 | 64.8 | 36.7 | 5 |
| PLM | PPITrans | 0.362 | 3.39 | 52.2 | 29.1 | 26.3 | 0.314 | 4.54 | 418 | 270 | 46.6 | 0.449 | 2.57 | 71.9 | 46.1 | 19.6 | 5 |
| | TUnA | 0.342 | 2.99 | 47.8 | 30.2 | 22.9 | 0.289 | 4.44 | 416 | 272 | 46.0 | 0.450 | 2.23 | 55.4 | 42.5 | 13.7 | 4 |
| | PLM-interact (35M) | 0.383 | 2.29 | 24.4 | 16.0 | 12.8 | 0.322 | 2.85 | 209 | 78.2 | 36.3 | 0.429 | 2.47 | 52.6 | 22.6 | 13.6 | 2 |
| | PLM-interact (650M) | 0.396 | 1.64 | 30.9 | 14.8 | 16.5 | 0.350 | 2.03 | 236 | 63.6 | 42.2 | 0.491 | 1.76 | 50.3 | 21.2 | 15.4 | 1 |
| Structure | Struct2Graph | 0.27 | 5.58 | 95.1 | 50.5 | 39.2 | 0.132 | 10.8 | 508 | 321 | 67.6 | 0.237 | 5.83 | 155 | 103 | 38.3 | 10 |
| | TAGPPI | 0.283 | 4.56 | 70.8 | 36.2 | 29.5 | 0.143 | 9.94 | 491 | 309 | 65.3 | 0.256 | 4.87 | 127 | 74.2 | 28.6 | 9 |
| | RF2-Lite | 0.227 | 0.524 | 43.8 | 31.7 | 34.9 | 0.243 | 1.10 | 226 | 39.3 | 61.8 | 0.171 | 0.514 | 81.7 | 73.9 | 60.9 | 7 |

#### G.1.1   Intra-species PPI Network Construction

Tab. 7 and Tab. 8 report the performance of all models on graph-level and binary classification metrics, respectively. A detailed analysis of the graph-level results is provided in Sec. 4.3.1.

Table 8: Binary classification results of intra-species (Human) test set. We use three color scales of blue to denote the **first**, second, and third best performance.

| Category | Model | Accuracy↑ | Precision↑ | Recall↑ | F1↑ | AUPR↑ | Avg. Rank |
|---|---|---|---|---|---|---|---|
| Seq. Sim. | SPRINT | 0.545 | 0.607 | 0.257 | 0.361 | 0.527 | 7 |
| Naive Sequence | PIPR | 0.559 | 0.557 | 0.581 | 0.568 | 0.565 | 5 |
| | D-SCRIPT | 0.505 | 0.597 | 0.104 | 0.144 | 0.539 | 10 |
| | Topsy-Turvy | 0.508 | 0.596 | 0.120 | 0.160 | 0.541 | 9 |
| PLM | PPITrans | 0.674 | 0.657 | 0.738 | 0.694 | 0.769 | 1 |
| | TUnA | 0.647 | 0.676 | 0.563 | 0.614 | 0.720 | 1 |
| | PLM-interact (35M) | 0.640 | 0.653 | 0.597 | 0.623 | 0.706 | 4 |
| | PLM-interact (650M) | 0.671 | 0.735 | 0.534 | 0.619 | 0.707 | 3 |
| Structure | Struct2Graph | 0.515 | 0.511 | 0.696 | 0.589 | 0.507 | 8 |
| | TAGPPI | 0.545 | 0.536 | 0.672 | 0.596 | 0.524 | 6 |

Overall, the model rankings based on binary classification metrics are broadly consistent with those based on graph-level metrics, with PLM-based approaches outperforming other model types. Notably, PPITrans and TUnA achieve the highest classification scores, while PLM-interact demonstrates superior performance in reconstructing the overall PPI network topology.

Table 9: Graph-level results of cross-species (Human to Arath) PPI network construction task. We use three color scales of blue to denote the **first**, second, and third best performance. **RD** closer to 1 is better.

| | Sampling Method | BFS | | | | | DFS | | | | | RW | | | | | Avg. Rank |
|---|---|---|---|---|---|---|---|---|---|---|---|---|---|---|---|---|---|
| Category | Model | GS↑ | RD→ | Deg.↓ | Clus.↓ | Spectral↓ | GS↑ | RD→ | Deg.↓ | Clus.↓ | Spectral↓ | GS↑ | RD→ | Deg.↓ | Clus.↓ | Spectral↓ | Rank |
| Seq. Sim. | SPRINT | 0.200 | 1.27 | 10.5 | 15.7 | 21.0 | 0.149 | 2.06 | 9.11 | 108 | 40.5 | 0.204 | 0.847 | 11.8 | 20.8 | 28.2 | 3 |
| Naive Sequence | PIPR | 0.211 | 5.23 | 101 | 67.4 | 33.9 | 0.0886 | 11.9 | 80.7 | 306 | 64.9 | 0.219 | 5.47 | 123 | 81.1 | 35.7 | 10 |
| | D-SCRIPT | 0.167 | 1.31 | 21.6 | 27.0 | 37.5 | 0.111 | 8.09 | 55.6 | 241 | 67.1 | 0.247 | 6.27 | 118 | 57.0 | 53.3 | 8 |
| | Topsy-Turvy | 0.163 | 1.91 | 20.6 | 20.5 | 15.6 | 0.153 | 2.32 | 13.0 | 123 | 61.3 | 0.267 | 3.90 | 60.5 | 33.0 | 30.6 | 4 |
| PLM | PPITrans | 0.330 | 4.97 | 59.6 | 47.7 | 32.5 | 0.250 | 6.62 | 62.0 | 252 | 54.9 | 0.423 | 2.79 | 45.6 | 28.9 | 28.8 | 6 |
| | TUnA | 0.310 | 4.47 | 62.9 | 45.3 | 26.1 | 0.242 | 6.41 | 68.6 | 248 | 48.6 | 0.382 | 2.91 | 58.7 | 30.2 | 22.6 | 5 |
| | PLM-interact (35M) | 0.304 | 2.50 | 23.8 | 13.5 | 20.2 | 0.227 | 4.26 | 35.4 | 124 | 34.6 | 0.339 | 1.78 | 13.5 | 7.49 | 16.3 | 2 |
| | PLM-interact (650M) | 0.364 | 1.59 | 25.8 | 13.7 | 19.2 | 0.276 | 2.38 | 37.0 | 108 | 32.7 | 0.445 | 1.26 | 22.7 | 12.6 | 19.2 | 1 |
| Structure | Struct2Graph | 0.241 | 7.10 | 118 | 76.4 | 37.7 | 0.130 | 14.2 | 83.9 | 349 | 69.8 | 0.242 | 5.38 | 109 | 60.9 | 30.9 | 11 |
| | TAGPPI | 0.258 | 5.89 | 87.7 | 55.5 | 29.5 | 0.145 | 12.8 | 82.0 | 328 | 64.0 | 0.265 | 4.24 | 79.0 | 45.1 | 22.4 | 8 |
| | RF2-Lite | 0.104 | 0.470 | 49.8 | 87.2 | 52.4 | 0.161 | 1.08 | 7.59 | 167 | 43.9 | 0.184 | 0.442 | 30.0 | 51.8 | 43.8 | 7 |

Table 10: Binary classification results of cross-species (Human to Arath) test set. We use three color scales of blue to denote the **first**, second, and third best performance.

| Category | Model | Accuracy↑ | Precision↑ | Recall↑ | F1↑ | AUPR↑ | Avg. Rank |
|---|---|---|---|---|---|---|---|
| Seq. Sim. | SPRINT | 0.552 | 0.702 | 0.181 | 0.287 | 0.537 | 7 |
| Naive Sequene | PIPR | 0.579 | 0.568 | 0.659 | 0.610 | 0.586 | 4 |
| | D-SCRIPT | 0.509 | 0.734 | 0.118 | 0.155 | 0.570 | 9 |
| | Topsy-Turvy | 0.501 | 0.696 | 0.118 | 0.158 | 0.556 | 10 |
| PLM | PPITrans | 0.748 | 0.757 | 0.731 | 0.744 | 0.827 | 1 |
| | TUnA | 0.702 | 0.758 | 0.593 | 0.665 | 0.780 | 2 |
| | PLM-interact (35M) | 0.621 | 0.706 | 0.413 | 0.521 | 0.677 | 5 |
| | PLM-interact (650M) | 0.661 | 0.811 | 0.420 | 0.554 | 0.757 | 3 |
| Structure | Struct2Graph | 0.507 | 0.505 | 0.671 | 0.576 | 0.503 | 7 |
| | TAGPPI | 0.522 | 0.518 | 0.655 | 0.578 | 0.512 | 6 |

Table 11: Graph-level results of cross-species (Human to Yeast) PPI network construction task. We use three color scales of blue to denote the **first**, second, and third best performance. **RD** closer to 1 is better.

| | Sampling Method | BFS | | | | | DFS | | | | | RW | | | | | Avg. Rank |
|---|---|---|---|---|---|---|---|---|---|---|---|---|---|---|---|---|---|
| Category | Model | GS↑ | RD→ | Deg.↓ | Clus.↓ | Spectral↓ | GS↑ | RD→ | Deg.↓ | Clus.↓ | Spectral↓ | GS↑ | RD→ | Deg.↓ | Clus.↓ | Spectral↓ | Rank |
| Seq. Sim. | SPRINT | 0.242 | 1.04 | 12.8 | 20.2 | 16.5 | 0.171 | 1.72 | 52.1 | 130 | 31.6 | 0.256 | 1.65 | 19.8 | 42.1 | 18.2 | 2 |
| Naive Sequence | PIPR | 0.196 | 4.35 | 90.0 | 49.9 | 30.2 | 0.0957 | 8.96 | 531 | 278 | 50.6 | 0.173 | 6.38 | 180 | 92.2 | 38.4 | 11 |
| | D-SCRIPT | 0.168 | 1.14 | 14.2 | 26.5 | 34.8 | 0.106 | 5.28 | 221 | 189 | 49.7 | 0.202 | 8.04 | 173 | 90.3 | 59.1 | 7 |
| | Topsy-Turvy | 0.142 | 1.55 | 11.6 | 14.2 | 10.9 | 0.130 | 2.58 | 115 | 101 | 28.5 | 0.186 | 5.12 | 100.0 | 50.1 | 35.0 | 4 |
| PLM | PPITrans | 0.348 | 4.43 | 55.4 | 35.1 | 30.8 | 0.200 | 8.15 | 397 | 257 | 49.0 | 0.339 | 4.44 | 95.3 | 50.1 | 35.0 | 6 |
| | TUnA | 0.332 | 3.60 | 57.7 | 37.8 | 22.6 | 0.219 | 5.47 | 414 | 225 | 36.1 | 0.323 | 4.07 | 119 | 56.0 | 26.3 | 5 |
| | PLM-interact (35M) | 0.314 | 2.34 | 24.0 | 12.5 | 15.5 | 0.189 | 3.95 | 202 | 119 | 29.4 | 0.305 | 2.82 | 49.7 | 21.3 | 17.4 | 3 |
| | PLM-interact (650M) | 0.354 | 1.90 | 25.8 | 12.7 | 14.1 | 0.258 | 3.04 | 232 | 85.8 | 27.9 | 0.365 | 2.10 | 51.3 | 19.4 | 13.8 | 1 |
| Structure | Struct2Graph | 0.256 | 5.89 | 115 | 68.4 | 37.1 | 0.139 | 11.6 | 546 | 337 | 55.8 | 0.211 | 6.87 | 178 | 97.7 | 41.1 | 9 |
| | TAGPPI | 0.279 | 4.82 | 91.7 | 58.8 | 28.8 | 0.154 | 10.1 | 519 | 311 | 51.2 | 0.230 | 5.67 | 151 | 78.6 | 31.6 | 8 |
| | RF2-Lite | 0.136 | 0.361 | 71.6 | 91.0 | 73.7 | 0.179 | 0.699 | 158 | 170 | 54.1 | 0.138 | 0.359 | 89.1 | 127 | 69.6 | 10 |

Table 12: Binary classification results of cross-species (Human to Yeast) test set. We use three color scales of blue to denote the **first**, **second**, and third best performance.

| Category | Model | Accuracy↑ | Precision↑ | Recall↑ | F1↑ | AUPR↑ | Avg. Rank |
|---|---|---|---|---|---|---|---|
| Seq. Sim. | SPRINT | 0.554 | 0.620 | 0.280 | 0.386 | 0.534 | 7 |
| Naive Sequene | PIPR | 0.552 | 0.551 | 0.558 | 0.555 | 0.563 | 5 |
| | D-SCRIPT | 0.516 | 0.554 | 0.167 | 0.256 | 0.523 | 9 |
| | Topsy-Turvy | 0.509 | 0.529 | 0.168 | 0.255 | 0.506 | 10 |
| PLM | PPITrans | 0.729 | 0.700 | 0.806 | 0.749 | 0.815 | 1 |
| | TUnA | 0.689 | 0.688 | 0.690 | 0.690 | 0.766 | 2 |
| | PLM-interact (35M) | 0.615 | 0.655 | 0.484 | 0.557 | 0.667 | 4 |
| | PLM-interact (650M) | 0.660 | 0.743 | 0.490 | 0.590 | 0.725 | 3 |
| Structure | Struct2Graph | 0.511 | 0.508 | 0.709 | 0.592 | 0.506 | 8 |
| | TAGPPI | 0.528 | 0.521 | 0.696 | 0.596 | 0.514 | 5 |

Table 13: Graph-level results of cross-species (Human to Ecoli) PPI network construction task. We use three color scales of blue to denote the **first**, **second**, and third best performance. **RD** closer to 1 is better.

| Sampling Method | | BFS | | | | | DFS | | | | | RW | | | | | Avg. |
|---|---|---|---|---|---|---|---|---|---|---|---|---|---|---|---|---|---|
| Category | Model | GS↑ | RD→ | Deg.↓ | Clus.↓ | Spectral↓ | GS↑ | RD→ | Deg.↓ | Clus.↓ | Spectral↓ | GS↑ | RD→ | Deg.↓ | Clus.↓ | Spectral↓ | Rank |
| Seq. Sim. | SPRINT | 0.147 | 0.573 | 51.6 | 37.5 | 41.7 | 0.125 | 0.860 | 164 | 116 | 55.2 | 0.157 | 0.765 | 100.0 | 57.9 | 44.1 | 2 |
| Naive Sequence | PIPR | 0.172 | 5.27 | 144 | 46.3 | 45.2 | 0.0814 | 11.6 | 780 | 359 | 63.1 | 0.134 | 8.57 | 437 | 129 | 58.0 | 11 |
| | D-SCRIPT | 0.159 | 1.78 | 20.6 | 30.5 | 40.2 | 0.108 | 10.1 | 624 | 286 | 58.8 | 0.145 | 10.3 | 410 | 119 | 75.6 | 7 |
| | Topsy-Turvy | 0.220 | 5.54 | 132 | 43.6 | 39.0 | 0.0810 | 3.26 | 312 | 126 | 50.1 | 0.160 | 10.8 | 422 | 152 | 70.4 | 8 |
| PLM | PPITrans | 0.298 | 4.94 | 105 | 40.8 | 39.4 | 0.158 | 9.27 | 711 | 347 | 54.8 | 0.243 | 5.65 | 327 | 92.0 | 45.5 | 4 |
| | TUnA | 0.271 | 5.80 | 127 | 51.2 | 43.2 | 0.144 | 10.9 | 664 | 345 | 56.9 | 0.204 | 8.17 | 392 | 125 | 57.0 | 8 |
| | PLM-interact (35M) | 0.251 | 3.68 | 83.8 | 26.7 | 24.9 | 0.151 | 5.68 | 459 | 218 | 37.8 | 0.201 | 4.85 | 273 | 67.5 | 33.2 | 1 |
| | PLM-interact (650M) | 0.267 | 4.88 | 113 | 38.5 | 33.6 | 0.152 | 7.65 | 589 | 275 | 44.1 | 0.215 | 6.31 | 326 | 89.5 | 42.6 | 3 |
| Structure | Struct2Graph | 0.210 | 5.22 | 144 | 54.1 | 37.7 | 0.0994 | 11.0 | 771 | 371 | 57.6 | 0.155 | 7.15 | 393 | 108 | 48.0 | 10 |
| | TAGPPI | 0.237 | 4.88 | 136 | 54.1 | 35.0 | 0.110 | 10.7 | 756 | 368 | 56.6 | 0.176 | 6.81 | 383 | 105 | 45.6 | 6 |
| | RF2-Lite | 0.152 | 0.395 | 116 | 86.7 | 78.9 | 0.197 | 0.585 | 273 | 252 | 77.0 | 0.166 | 0.388 | 225 | 131 | 91.7 | 5 |

Table 14: Binary classification results of cross-species (Human to Ecoli) test set. We use three color scales of blue to denote the **first**, **second**, and third best performance.

| Category | Model | Accuracy | Precision | Recall | F1 | AUPR | Avg. Rank |
|---|---|---|---|---|---|---|---|
| Seq. Sim. | SPRINT | 0.521 | 0.747 | 0.063 | 0.116 | 0.516 | 7 |
| Naive Sequene | PIPR | 0.573 | 0.573 | 0.568 | 0.571 | 0.578 | 5 |
| | D-SCRIPT | 0.509 | 0.703 | 0.0782 | 0.120 | 0.548 | 6 |
| | Topsy-Turvy | 0.507 | 0.538 | 0.211 | 0.219 | 0.539 | 9 |
| PLM | PPITrans | 0.617 | 0.591 | 0.820 | 0.682 | 0.718 | 1 |
| | TUnA | 0.624 | 0.598 | 0.756 | 0.667 | 0.675 | 1 |
| | PLM-interact (35M) | 0.580 | 0.593 | 0.507 | 0.547 | 0.610 | 4 |
| | PLM-interact (650M) | 0.609 | 0.599 | 0.660 | 0.628 | 0.638 | 3 |
| Structure | Struct2Graph | 0.500 | 0.500 | 0.543 | 0.520 | 0.500 | 10 |
| | TAGPPI | 0.516 | 0.514 | 0.581 | 0.546 | 0.508 | 7 |

### G.1.2 Cross-species PPI network Construction

We report detailed experimental results for both graph-level and binary classification metrics in the cross-species evaluation setting.

Specifically,

- Tab. 9 and Tab. 10 present the results on **Arath**,
- Tab. 11 and Tab. 12 report the performance on **Yeast**, and
- Tab. 13 and Tab. 14 summarize the results on **Ecoli**.

These tables collectively illustrate how well models generalize to phylogenetically diverse species beyond the human training data.

We further account for numerical phylogenetic distance to provide a deeper understanding of cross-species generalization. To investigate this, we conducted an ablation study by computing the Pearson correlation between model performance and the estimated phylogenetic distance from Human. Specifically, we set the intra-species (Human→Human) distance to 0, and adopt widely accepted phylogenetic estimates [120] for other species: Human→Arath (1.5 Gyr), Human→Yeast (1.2 Gyr), and Human→Ecoli (3.6 Gyr), which consist the distance vector $\mathbf{d} = [0, 1.5, 1.3, 1.6]$. For given model $M$ and performance metric (*e.g.*, graph similarity (GS)), let $\mathbf{p}_m^M = [p_{\text{Human}}, p_{\text{Arath}}, p_{\text{Yeast}}, p_{\text{Ecoli}}]$

Table 15: Pearson correlation between phylogenetic distance and graph-level metrics.

| Sampling Method | | BFS | | | | | DFS | | | | | Random Walk | | | | |
|---|---|---|---|---|---|---|---|---|---|---|---|---|---|---|---|---|
| Category | Method | $r(\mathbf{d}, \mathbf{p}_{\text{GS}})$ | $r(\mathbf{d}, \mathbf{p}_{\text{RD}})$ | $r(\mathbf{d}, \mathbf{p}_{\text{Deg.}})$ | $r(\mathbf{d}, \mathbf{p}_{\text{Clus.}})$ | $r(\mathbf{d}, \mathbf{p}_{\text{Spectral}})$ | $r(\mathbf{d}, \mathbf{p}_{\text{GS}})$ | $r(\mathbf{d}, \mathbf{p}_{\text{RD}})$ | $r(\mathbf{d}, \mathbf{p}_{\text{Deg.}})$ | $r(\mathbf{d}, \mathbf{p}_{\text{Clus.}})$ | $r(\mathbf{d}, \mathbf{p}_{\text{Spectral}})$ | $r(\mathbf{d}, \mathbf{p}_{\text{GS}})$ | $r(\mathbf{d}, \mathbf{p}_{\text{RD}})$ | $r(\mathbf{d}, \mathbf{p}_{\text{Deg.}})$ | $r(\mathbf{d}, \mathbf{p}_{\text{Clus.}})$ | $r(\mathbf{d}, \mathbf{p}_{\text{Spectral}})$ |
| Naive Seq. | D-SCRIPT | -0.82 | 0.94 | 0.61 | 0.92 | 0.93 | -0.70 | 0.81 | 0.65 | 0.63 | -0.30 | -0.69 | 0.91 | 0.89 | 0.81 | 0.93 |
| PLM | PPITrans | -0.98 | 0.80 | 0.94 | 0.56 | 1.00 | -0.90 | 0.89 | 0.53 | 0.81 | 0.79 | -0.91 | 0.86 | 0.87 | 0.77 | 0.95 |
| | TUnA | -0.97 | 0.97 | 0.96 | 0.94 | 0.92 | -0.98 | 0.98 | 0.49 | 0.70 | 0.64 | -0.96 | 0.95 | 0.92 | 0.86 | 0.97 |
| Structure | Struct2Graph | -0.98 | -0.26 | 1.00 | -0.01 | -0.55 | -0.85 | -0.03 | 0.43 | 0.98 | -0.53 | -0.88 | 0.63 | 0.84 | 0.17 | 0.61 |

denotes the model's performance vector across the four species. We calculate the Pearson correlation $r(\mathbf{d}, \mathbf{p}_m^M)$ to quantify the relationship between phylogenetic distance and model performance. This analysis is conducted on four representative baselines, and are summarized in Tab. 15.

We find that phylogenetic distance is strongly correlated with model performance degradation across species. For all models and sampling strategies, GS consistently exhibits a strong negative correlation with phylogenetic distance (correlations typically < -0.7). This indicates that as phylogenetic distance increases, GS decreases—*i.e.*, model predictions become less similar to the ground-truth network. Most other metrics show strong positive correlations with phylogenetic distance. This means that as the evolutionary gap widens, predicted networks increasingly diverge from true biological networks, reflected by higher RD, degree, clustering, or spectral values, all of which signal degraded performance.

Table 16: Alternative Cross-species Transfer.

| Transfer Direction | | Arath→Arath | | | | | Arath→Yeast | | | | | Arath→Ecoli | | | | |
|---|---|---|---|---|---|---|---|---|---|---|---|---|---|---|---|---|
| Category | Method | GS↑ | RD→ | Deg.↓ | Clus.↓ | Spectral↓ | GS↑ | RD→ | Deg.↓ | Clus.↓ | Spectral↓ | GS↑ | RD→ | Deg.↓ | Clus.↓ | Spectral↓ |
| Naive Seq. | D-SCRIPT | 0.407 | 2.24 | 38.4 | 11.6 | 19.0 | 0.288 | 3.68 | 52.3 | 25.0 | 28.9 | 0.243 | 6.11 | 121 | 46.1 | 44.1 |
| PLM | PPITrans | 0.470 | 2.96 | 43.7 | 31.4 | 17.8 | 0.376 | 4.68 | 78.0 | 50.8 | 35.8 | 0.312 | 5.14 | 138 | 51.4 | 44.1 |
| | TUnA | 0.548 | 1.54 | 22.6 | 19.1 | 14.4 | 0.378 | 3.87 | 58.5 | 42.9 | 27.0 | 0.302 | 5.39 | 129 | 49.9 | 42.1 |
| Structure | Struct2Graph | 0.352 | 2.59 | 67.0 | 70.5 | 16.3 | 0.266 | 4.84 | 92.9 | 62.1 | 30.6 | 0.218 | 5.53 | 145 | 61 | 38.8 |

In addition to the cross-species transfer direction presented in the main manuscript, we perform new experiments on botany-to-fungi (Arath → Yeast) and botany-to-bacteria (Arath → Ecoli) transfer learning. We followed the same data construction pipeline described in Sec. 3.1 to avoid data leakage. We focused on the BFS sampling strategy and evaluated four representative models. The results are presented in Tab. 16, and they confirm our previous experimental observation: cross-species generalization becomes more challenging with greater evolutionary divergence.

Specifically, compared to the intra-species setting (Arath → Arath), the GS scores for Arath → Yeast and Arath → Ecoli decreased by 26% and 39%, respectively. Other graph-level metrics observe performance degradation as well, indicating the performance of the transfer learning decreases with the increased evolutionary distances.

### G.1.3 Correlation Between Binary Classification Scores and Network Metrics

To investigate whether standard classification metrics reflect the structural quality of reconstructed PPI networks, we analyze their correlation with five topology-aware metrics across all evaluated models. We summarize some important discoveries in Fig. 8:

- **Recall often distorts the network topology**: Recall measures the proportion of true positives correctly identified. To achieve higher recall, the model typically predicts more edges to capture as many true interactions as possible, even at the cost of including false positives. This leads to a denser predicted graph, which deviates from the sparsity characteristic of real PPI networks, ultimately hurting the preservation of the original topological structure.

- **Precision tends to preserve network topology**: Precision measures the proportion of correct positive predictions. Improving precision requires minimizing false positives, which makes the model more conservative in predicting edges. As a result, the predicted graph becomes sparser and more aligned with the intrinsic sparsity of real PPI networks, thereby helping to preserve the underlying network topology.

- **Composite metrics like F1-score and AUPR may hide topological differences**: Since these metrics balance both precision and recall, they may yield high scores even when the predicted network substantially deviates from the true topology. As a result, their correlation with topological fidelity tends to be weak, potentially masking structural distortions in the predicted network.

These results underscore the importance of evaluating PPI prediction models beyond binary classification, as traditional metrics alone may fail to reflect structural coherence and biological relevance.

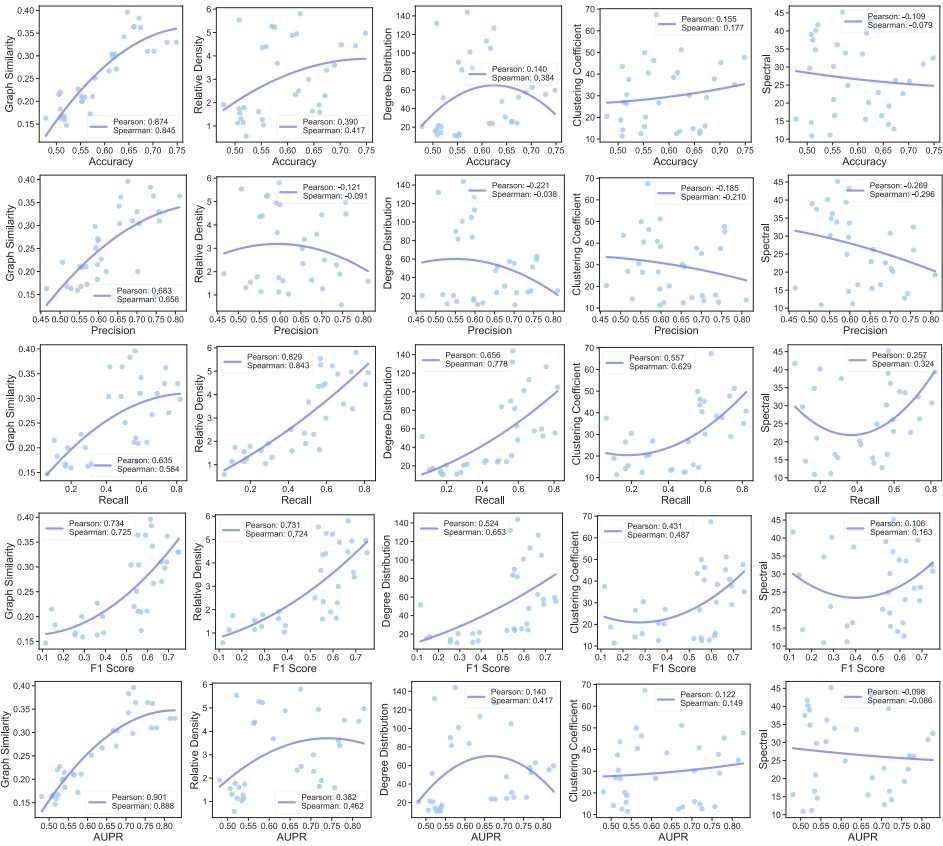

Figure 8: Correlation between binary classification metrics and graph-level topology metrics. Each subplot shows the Pearson and Spearman correlation between a classification metric and a graph-level metric. Lastly, a second-order polynomial regression is fit to the data to capture non-linear trends.

### G.1.4 Impact of Subgraph Size on Topology-Oriented Performance

In the test subgraphs, as previously described, the node size ranges from 20 to 200. To investigate the impact of node size on model performance, we visualize the results in a bar plot, as shown in Fig. 9. We select PLM-interact (650M), the best-performing model overall, to assess how topological fidelity changes with subgraph scale. As node size increases, we observe a gradual decline in Graph Similarity, suggesting that reconstructing global structures becomes more challenging in larger subgraphs. Conversely, Relative Density, Clustering Coefficient (MMD), Degree Distribution (MMD), and Spectral (MMD) generally exhibit increasing trends, which indicate degraded alignment with the ground truth in these structural properties. These results suggest that larger PPI networks may exhibit more complex topological patterns, posing greater challenges for faithful reconstruction.

Table 17: Results for protein complex pathway prediction task. We use three color scales of blue to denote the first, second, and third best performance.

| Category | Model | Pathway Recall↑ | Pathway Precision↑ | Pathway Connectivity↑ | Avg. Rank |
|---|---|---|---|---|---|
| Seq. Sim. | SPRINT | 0.526 | 0.856 | 0.752 | 6 |
| Naive Sequene | PIPR | 0.160 | 0.588 | 0.323 | 10 |
| | D-SCRIPT | 0.225 | 0.466 | 0.316 | 11 |
| | Topsy-Turvy | 0.240 | 0.613 | 0.338 | 9 |
| PLM | PPITrans | 0.583 | 0.863 | 0.818 | 1 |
| | TUnA | 0.526 | 0.864 | 0.794 | 3 |
| | PLM-interact (35M) | 0.550 | 0.847 | 0.880 | 3 |
| | PLM-interact (650M) | 0.549 | 0.862 | 0.887 | 2 |
| Structure | Struct2Graph | 0.413 | 0.742 | 0.608 | 8 |
| | TAGPPI | 0.487 | 0.781 | 0.713 | 7 |
| | RF2-Lite | 0.554 | 0.874 | 0.135 | 5 |

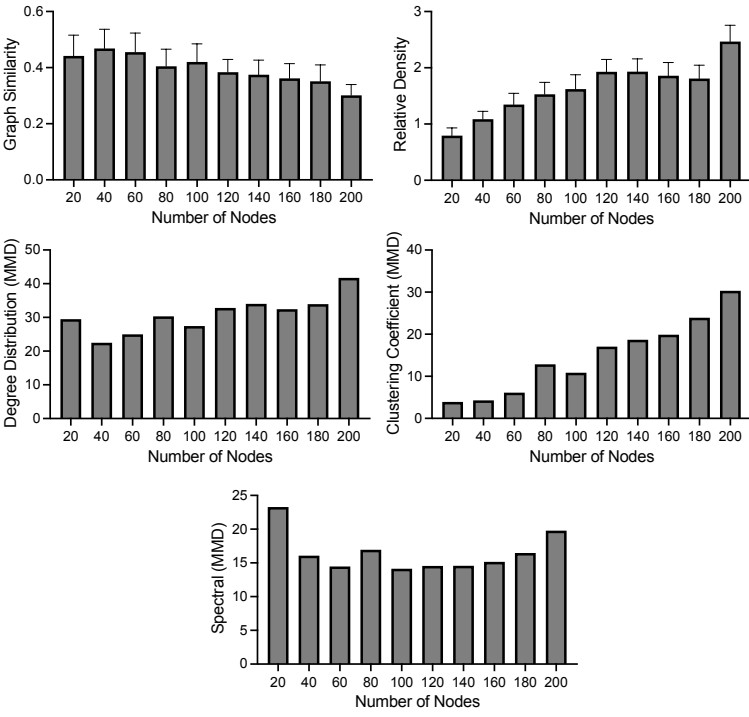

Figure 9: Effect of subgraph node size on graph-level metrics. The error bars indicate the 95% confidence intervals.

Table 18: Results for GO enrichment analysis task. We use three color scales of blue to denote the **first**, **second**, and **third** best performance.

| Category | | | GO:BP | | GO:MF | | GO:CC | | Average | | |
|---|---|---|---|---|---|---|---|---|---|---|---|
| **Category** | **Model** | | **FA↑** | **CR↑** | **FA↑** | **CR↑** | **FA↑** | **CR↑** | **FA↑** | **CR↑** | **Avg. Rank** |
| Seq. Sim. | SPRINT | | 0.174 | 0.468 | 0.094 | 0.660 | 0.254 | 0.838 | 0.174 | 0.655 | 7 |
| Naive Sequene | PIPR | | 0.153 | 0.480 | 0.145 | 0.656 | 0.247 | 0.827 | 0.182 | 0.654 | 4 |
| | D-SCRIPT | | 0.125 | 0.462 | 0.113 | 0.668 | 0.286 | 0.904 | 0.175 | 0.678 | 4 |
| | Topsy-Turvy | | 0.131 | 0.333 | 0.242 | 0.652 | 0.323 | 0.919 | 0.232 | 0.635 | 7 |
| PLM | PPITrans | | 0.183 | 0.632 | 0.236 | 0.717 | 0.330 | 0.919 | 0.250 | 0.756 | 3 |
| | TUnA | | 0.160 | 0.404 | 0.223 | 0.595 | 0.395 | 0.868 | 0.259 | 0.622 | 6 |
| | PLM-interact (35M) | | 0.264 | 0.684 | 0.322 | 0.713 | 0.419 | 0.959 | 0.335 | 0.785 | 2 |
| | PLM-interact (650M) | | 0.337 | 0.789 | 0.359 | 0.773 | 0.408 | 0.970 | 0.368 | 0.844 | 1 |
| Structure | Struct2Graph | | 0.150 | 0.469 | 0.106 | 0.653 | 0.207 | 0.826 | 0.154 | 0.649 | 10 |
| | TAGPPI | | 0.180 | 0.468 | 0.111 | 0.653 | 0.212 | 0.826 | 0.168 | 0.649 | 9 |

## G.2 Function-oriented Task

### G.2.1 Protein Complex Pathway Prediction

As shown in Tab. 17, PLM-based models outperform others, with PPITrans achieving the best overall performance. PLM-interact (650M) ranks second, excelling in pathway connectivity. While RF2-Lite shows strong precision, its low connectivity limits overall ranking. Among non-PLM methods, only SPRINT performs competitively, indicating that PLMs better capture complex-level functional organization.

### G.2.2 GO Enrichment Analysis

In addition to the average scores across the three GO ontologies, Biological Process (BP), Molecular Function (MF), and Cellular Component (CC), we report detailed results for each category in Tab. 18. PLM-interact (650M) consistently achieves the highest performance across all GO categories, demonstrating strong functional coherence in its predicted networks. PLM-interact (35M) and PPITrans also perform competitively. Nevertheless, the low function alignment scores observed

across all models suggest that current PPI prediction methods still struggle to capture fine-grained functional modules, indicating room for further improvement.

Table 19: Results for essential protein justification task. We use three color scales of blue to denote the **first**, second, and third best performance.

| Category | Model | Precision@100↑ | Distribution Overlap↓ | Avg. Rank |
|---|---|---|---|---|
| Seq. Sim. | SPRINT | 0.54 | 0.727 | 3 |
| Naive Sequence | PIPR | 0.52 | 0.834 | 5 |
| | D-SCRIPT | 0.47 | 0.629 | 4 |
| | Topsy-Turvy | 0.43 | 0.755 | 7 |
| PLM | PPITrans | 0.48 | 0.886 | 9 |
| | TUnA | 0.57 | 0.849 | 5 |
| | PLM-interact (35M) | 0.8 | 0.423 | 1 |
| | PLM-interact (650M) | 0.77 | 0.440 | 2 |
| Structure | Struct2Graph | 0.45 | 0.841 | 9 |
| | TAGPPI | 0.43 | 0.828 | 8 |

### G.2.3 Essential Protein Justification

The more detailed results analysis for the essential protein justification task is given in Tab. 19. Again, the PLM-interact series achieves the best performance Nevertheless, it should be noted that all baseline methods exhibit a relatively large distribution overlap between essential and non-essential proteins in the reconstructed PPI networks. This reflects the limited ability of current PPI prediction models to preserve functional properties such as node centrality, which may hinder their effectiveness in supporting downstream biological applications, including drug discovery and disease gene prioritization. In Fig. 10, we visualize the network centrality distributions for essential and non-essential proteins across all baseline models and compare them to the ground truth. The results further highlight the need to improve models' ability to preserve centrality signals to accurately identify essential proteins.

Table 20: Ablation results on the positive-to-negative ratio of training pairs under three sampling configurations.

| | | BFS | | | | | | | | | | | | | | |
|---|---|---|---|---|---|---|---|---|---|---|---|---|---|---|---|---|
| **Sampling Method** | | **Ratio 1:1** | | | | | **Ratio 1:5** | | | | | **Ratio 1:10** | | | | |
| Category | Model | GS↑ | RD→ | Deg.↓ | Clus.↓ | Spectral↓ | GS↑ | RD→ | Deg.↓ | Clus.↓ | Spectral↓ | GS↑ | RD→ | Deg.↓ | Clus.↓ | Spectral↓ |
| Naive Sequence | D-SCRIPT | 0.215 | 1.13 | 15.4 | 18.8 | 29.7 | 0.032 | 0.0705 | 94.2 | 46.6 | 83.2 | 0.035 | 0.0400 | 116 | 54.1 | 97.2 |
| PLM | PPITrans | 0.362 | 3.39 | 52.2 | 29.1 | 26.3 | 0.393 | 1.17 | 8.12 | 7.37 | 10.9 | 0.374 | 0.692 | 11.3 | 7.23 | 16.4 |
| | TUnA | 0.342 | 2.99 | 47.8 | 30.2 | 22.9 | 0.346 | 1.14 | 10.7 | 9.06 | 8.77 | 0.314 | 0.579 | 14.4 | 9.58 | 17.3 |
| Structure | Struct2Graph | 0.270 | 5.58 | 95.1 | 50.5 | 39.2 | 0.236 | 4.11 | 62.6 | 35.2 | 25.6 | 0 | 5.71E-05 | 146 | 65.8 | 117 |
| | | **DFS** | | | | | | | | | | | | | | |
| **Sampling Method** | | **Ratio 1:1** | | | | | **Ratio 1:5** | | | | | **Ratio 1:10** | | | | |
| Category | Model | GS↑ | RD→ | Deg.↓ | Clus.↓ | Spectral↓ | GS↑ | RD→ | Deg.↓ | Clus.↓ | Spectral↓ | GS↑ | RD→ | Deg.↓ | Clus.↓ | Spectral↓ |
| Naive Sequence | D-SCRIPT | 0.123 | 6.40 | 302 | 237 | 67.0 | 0.0649 | 0.0545 | 518 | 137 | 136 | 0.089 | 0.27 | 413 | 112 | 107 |
| PLM | PPITrans | 0.314 | 4.54 | 418 | 270 | 46.6 | 0.433 | 2.19 | 213 | 153 | 28.3 | 0.520 | 1.19 | 100 | 40.0 | 38.0 |
| | TUnA | 0.289 | 4.44 | 416 | 272 | 46.0 | 0.417 | 1.49 | 127 | 85.6 | 19.7 | 0.414 | 0.933 | 114 | 21.9 | 38.9 |
| Structure | Struct2Graph | 0.132 | 10.8 | 508 | 321 | 67.6 | 0.118 | 7.29 | 430 | 282 | 50.9 | 1.74E-05 | 3.50E-05 | 593 | 154 | 145 |
| | | **Random Walk** | | | | | | | | | | | | | | |
| **Sampling Method** | | **Ratio 1:1** | | | | | **Ratio 1:5** | | | | | **Ratio 1:10** | | | | |
| Category | Model | GS↑ | RD→ | Deg.↓ | Clus.↓ | Spectral↓ | GS↑ | RD→ | Deg.↓ | Clus.↓ | Spectral↓ | GS↑ | RD→ | Deg.↓ | Clus.↓ | Spectral↓ |
| Naive Sequence | D-SCRIPT | 0.209 | 5.82 | 134 | 64.8 | 52.0 | 0.135 | 0.334 | 83.2 | 51.9 | 55.6 | 0.01 | 0.01 | 188 | 133 | 134 |
| PLM | PPITrans | 0.449 | 2.57 | 71.9 | 46.1 | 19.6 | 0.564 | 1.27 | 10.8 | 10.8 | 12.3 | 0.547 | 0.808 | 23.7 | 14.9 | 25 |
| | TUnA | 0.450 | 2.23 | 55.4 | 42.5 | 13.7 | 0.463 | 0.764 | 15.8 | 9.58 | 13.2 | 0.425 | 0.539 | 38.0 | 27.6 | 29.7 |
| Structure | Struct2Graph | 0.237 | 5.83 | 155 | 103 | 38.3 | 0.216 | 3.44 | 87.7 | 72.0 | 18.4 | 0 | 5.71E-05 | 197 | 135 | 140 |

### G.3 Ablation Study on Positive-to-Negative Ratio

To examine the effect of class imbalance on PPI network reconstruction, we vary the positive-to-negative ratio of training pairs from 1:1 to 1:10 under three sampling configurations: BFS, DFS, and Random Walk. This setting reflects the inherent sparsity of real PPI networks, where non-interacting protein pairs largely outnumber interacting ones. We evaluate representative sequence-, PLM-, and structure-based models using five network-level metrics.

Tab. 20 reports the complete results. Across sampling strategies, moderate imbalance (*e.g.*, 1:5) tends to improve the robustness of PLM-based models (PPITrans, TUnA), while excessive imbalance (1:10)

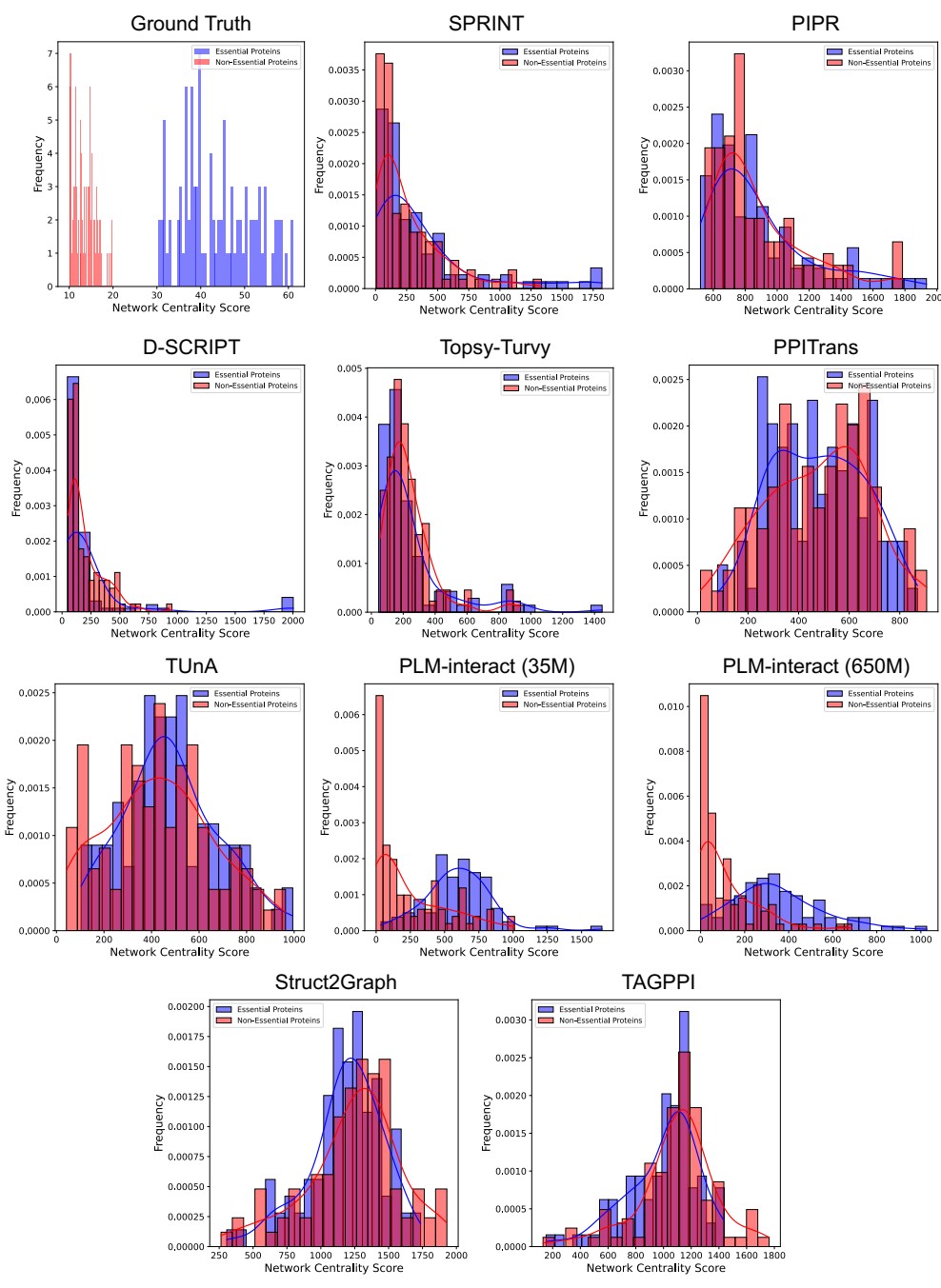

Figure 10: Network centrality distributions of essential and non-essential proteins.

causes performance degradation and overly sparse reconstructions. In contrast, simpler models such as D-SCRIPT and Struct2Graph deteriorate rapidly as the ratio increases, indicating their sensitivity to biased supervision.

## G.4 Ablation Study on Probability Threshold

We evaluate how varying probability thresholds influence the balance between precision and recall across four representative models. As shown in Tab. 21, Struct2Graph and D-SCRIPT are highly sensitive to thresholding, with recall dropping sharply, indicating that a large portion of their prediction

Table 21: Effect of Probability Thresholds on Precision and Recall.

| | PIPR | | D-SCRIPT | | PPITrans | | Struct2Graph | |
|---|---|---|---|---|---|---|---|---|
| Threshold | Precision | Recall | Precision | Recall | Precision | Recall | Precision | Recall |
| 0.2 | 0.552 | 0.634 | 0.570 | 0.135 | 0.627 | 0.821 | 0.502 | 0.937 |
| 0.3 | 0.554 | 0.615 | 0.582 | 0.115 | 0.651 | 0.769 | 0.504 | 0.900 |
| 0.5 | 0.557 | 0.581 | 0.597 | 0.104 | 0.658 | 0.738 | 0.511 | 0.696 |
| 0.7 | 0.561 | 0.550 | 0.764 | 0.067 | 0.753 | 0.535 | 0.532 | 0.007 |
| 0.8 | 0.562 | 0.530 | 0.780 | 0.053 | 0.798 | 0.438 | 0.532 | 0.007 |

scores have low confidence and are clustered around 0.5. In contrast, PIPR and PPITrans exhibit more stable trends: recall gradually decreases while precision increases as the threshold rises, suggesting a trade-off between the two metrics and a limited ability to maintain a strong balance.

### G.5 Case Study

We aim to assess the effectiveness of Chai-1 in reconstructing PPI networks. Owing to computational constraints, we select three representative graphs from the Human species, with node sizes ranging from 20 to 60. The detailed experimental setup is provided in Apx. F, and the corresponding results are summarized in Tab. 22.

The results reveal three key observations: (1) Chai-1 fails to achieve satisfactory performance in reconstructing PPI networks, as evidenced by a GS score of only 0.263 on a graph with 60 nodes; (2) the performance decreases as the graph size increases—for instance, the GS score drops from 0.425 at 20 nodes to 0.263 at 60 nodes—highlighting the challenge posed by larger topological scales; and (3) Chai-1 typically overpredicts interactions, resulting in high false positive rates, as reflected by an RD score of 6.14 on the 60-node graph, which is consistent with prior findings [70].

Furthermore, we visualize the predictions of Chai-1 in Fig. 11 to support our observations: while the ground-truth PPI networks are generally sparse, Chai-1's predictions are considerably denser.

Table 22: Graph-eval results of the Chai-1 on three graphs. **RD** closer to 1 is better.

| Node Size | | | 20 | | | | | 40 | | | | | 60 | | | |
|---|---|---|---|---|---|---|---|---|---|---|---|---|---|---|---|---|
| Category | Model | GS↑ | RD→ | Deg.↓ | Clus.↓ | Spectral↓ | GS↑ | RD→ | Deg.↓ | Clus.↓ | Spectral↓ | GS↑ | RD→ | Deg.↓ | Clus.↓ | Spectral↓ |
| Structure | Chai-1 | 0.425 | 3.33 | 0.522 | 0.338 | 0.574 | 0.346 | 4.380 | 0.449 | 0.384 | 0.563 | 0.263 | 6.14 | 0.787 | 0.666 | 0.747 |

### G.6 Scaling Analysis

Fig. 12 illustrates the graph similarity score against the size of the PPI prediction models. Overall, larger models tend to achieve better network reconstruction performance. The best model, PLM-interact (650M), reaches a graph similarity score of 0.41. Some structure-based models, such as Struct2Graph and TAGPPI, have comparable sizes to PLM-based models but fall much behind, achieving only half of the performance. This highlights that biologically-informed protein representations learned by PLMs play a crucial role in accurate PPI network reconstruction.

Nevertheless, the performance gains from increasing model size are relatively modest, suggesting that sheer model capacity alone is insufficient. Further improvements may require enhanced training objectives, or integration of complementary biological priors.

## H Broader Societal Impacts

This work introduces PRING, a comprehensive benchmark for PPI prediction, designed to advance evaluation from pairwise classification toward biologically grounded, network-level assessment. By offering a unified suite of topology-oriented and function-oriented tasks across multiple model organisms, PRING facilitates rigorous evaluation of a model's ability to reconstruct the structural topology of PPI networks, preserve functional coherence within biological modules, identify essential proteins, and recover meaningful functional pathways. This enables more faithful modeling of cellular systems and supports applications in systems biology, disease mechanism discovery, and therapeutic target identification. Moreover, our empirical analysis highlights the limitations of current computational approaches, revealing a gap between predictive accuracy and biological utility.

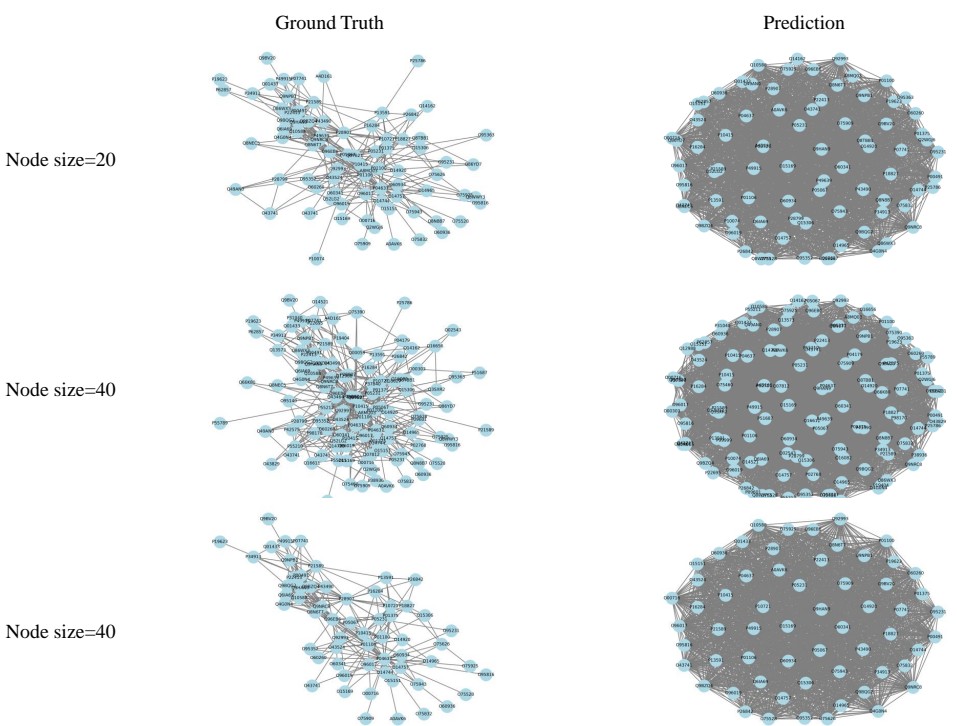

Figure 11: Visualization of Chai-1 predictions on three subgraphs with node sizes between 20 and 60.

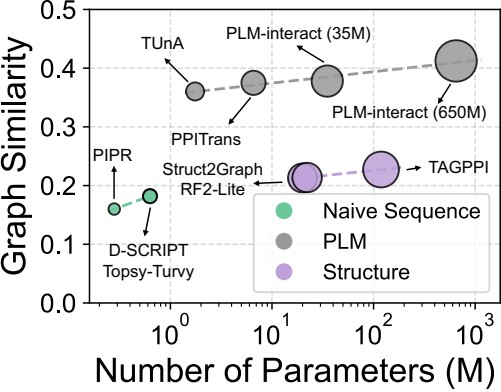

Figure 12: Scaling analysis of graph similarity.

Bridging this gap is essential for the responsible deployment of AI models in biomedical research and for accelerating foundational discoveries in life sciences.

Nevertheless, the potential risks associated with such dual-use scenarios are non-negligible. As PPI prediction models become more accurate and scalable, partly enabled by benchmarks like PRING, they may inadvertently lower the barrier for malicious actors to rationally design harmful biological agents. For example, enhanced understanding of host-pathogen interaction networks could be exploited to engineer synthetic pathogens that selectively disrupt immune functions or hijack critical cellular pathways [121, 122]. These scenarios, though speculative, underscore the need for vigilance in how such tools are disseminated and applied. Moving forward, it is essential to develop community norms and safeguards that promote transparency, ethical use, and oversight. This includes clear documentation of limitations, appropriate licensing, and collaboration with biosecurity experts to ensure that scientific progress does not come at the expense of public safety.

