# OpenReview forum: "PRING: Rethinking Protein-Protein Interaction Prediction from Pairs to Graphs"
_NeurIPS.cc/2025/Datasets_and_Benchmarks_Track — NeurIPS 2025 Datasets and Benchmarks Track poster_

### Official Review · Reviewer_bqYf · 2025-06-22

**Rating:** 5
**Confidence:** 4

**Summary:**

This manuscript proposes a brand new graph-based PPI benchmark. It integrates multiple PPI databases and spans multiple species. The authors propose two types of validation paradigms: topology-oriented tasks, including intra and cross-species PPI network construction, as well as function-oriented tasks, including pathway prediction, GO module analysis, and enssential protein justification.

**Additional Feedback:**

N/A

**Dataset Code Accessibility:**

Yes

**Dataset Code Comments:**

The dataset files are easy to obtain and have a clear format. It is recommended that the data format in each file be displayed more clearly.

**Ethical Considerations:**

No, there are no or only very minor ethics concerns

**Final Justification:**

As discussed, I have decided to keep my positive score.

**Limitations Weaknesses:**

- The reviewers' main concern with this manuscript is that each PPI dataset has unique and complex interaction construction rules. The reviewers expect the authors to propose some exciting ideas for merging these differentiated datasets. However, the merger ideas proposed in this manuscript seem rather simple and do not take the aforementioned issues into consideration.

- The statistics on PPI intra- and cross-species could be presented in greater detail. For example, how many proteins or PPIs are shared by different species (perhaps my thinking is not rigorous enough).

- Compared with many existing PPI network datasets, the reviewers felt that the PPI network proposed in the manuscript lacked obvious innovation. Intuitively, existing PPI datasets also seem to be able to achieve the graph evaluations.

**Strengths Contributions:**

- Compared with related work, this paper integrates PPI networks from a graphical perspective for the first time to form a benchmark. This perspective may be novel.
- This dataset effectively addresses the issues of data leakage and redundancy.
- The authors conducted thorough experimental verification and objectively clarified the challenges that exist in current PPI network research. And they claim that this benchmark represents positive progress toward a solution.

---

> ### Author Rebuttal · Authors · 2025-07-31
>
> We sincerely thank the reviewer for the thoughtful and constructive feedback. We are encouraged that the reviewer recognized the novelty of our graph-centric perspective, the rigorous data processing to avoid leakage and redundancy, and the comprehensive evaluation across multiple model categories. Meanwhile, we have carefully considered the weaknesses raised and provided detailed responses to each point below.
>
> > W1: The reviewers' main concern with this manuscript is that each PPI dataset has unique and complex interaction construction rules. The reviewers expect the authors to propose some exciting ideas for merging these differentiated datasets. However, the merger ideas proposed in this manuscript seem rather simple and do not take the aforementioned issues into consideration.
>
> **Response:** We appreciate the reviewer for raising this concern. We would like to respond from two perspectives: **(1) our current merging approach is rational and ensures high-quality and large-scale**, and **(2) we acknowledge the value and challenges in developing a novel merging strategy**.
>
> **From the first perspective**, our current design carefully considers the strengths of each data source to ensure both high data quality and large-scale coverage, without requiring additional filtering:
> 1. **IntAct** provides experimentally validated PPIs, which are widely regarded as high-confidence interactions.
> 2. **Reactome** contains PPIs supported by biological context, such as curated pathways and protein complexes, ensuring that the interactions have functional significance.
> 3. **UniProt** interactions are manually curated and supported by literature evidence, ensuring data reliability.
> 4. For **STRING**, we strictly follow the recommendations from the official documentation [1], also the common strategy used within the community, selecting only those interactions with a combined score ≥ 0.7 to ensure high qulaity.
>
> Our simple merging strategy achieves a balance between broad coverage and stringent quality control. As such, the resulting dataset is both comprehensive and reliable, aligning with our goal in *PRING* to enable fair and thorough evaluation of PPI network construction.
>
> **From the second perspective**, we recognize the potential value in more advanced, rule-based merging strategies, especially for scaling beyond the four databases we used. For example, in [2], the authors proposed a voting strategy to merge the PPIs from different data sources. While promising, such approaches may unintentionally discard interactions that are uniquely present in one high-quality source, reducing the completeness of the final dataset. Moreover, many databases provide only minimal metadata (e.g., protein IDs without contextual annotations), which limits the feasibility of designing a fully systematic merging protocol at scale.
>
> We believe that developing a novel merging framework is a meaningful direction, but it likely requires broader consensus and collaboration across the community. We will consider more promising merging strategies in our future work.
>
> > W2: The statistics on PPI intra- and cross-species could be presented in greater detail. For example, how many proteins or PPIs are shared by different species (perhaps my thinking is not rigorous enough).
>
> **Response:** We thank the reviewer for the helpful suggestion and agree that providing more detailed statistics on intra- and cross-species PPIs would enhance the clarity and completeness of the dataset description.
>
> First, we would like to clarify that there are no shared proteins or PPIs across the four species in the PRING dataset. This is a result of our rigorous filtering process: we applied both sequence similarity filtering (≤40%) using MMSeqs2 and gene function filtering to remove proteins with high similarity or identical functions across species. These steps were taken to ensure a clean separation between species-specific interaction networks, which is especially important for fair and unbiased cross-species evaluation.
>
> We have provided some statistics on *PRING* in Appendix C.3, including protein sequence lengths, number of proteins, graph node degrees, and the number of PPIs. Here, to further improve transparency, we now provide additional statistics detailing the impact of each filtering step. After retrieving PPI data from four curated databases, we first applied the protein filter (Fig. 2) to retain only experimentally verified proteins with known functions (from SwissProt) and belonging to one of the four target species. This step yielded: 15,043 Human proteins, 13,271 Arath proteins, 5,232 Yeast proteins, and 3,855 Ecoli proteins.
>
> Next, we applied sequence similarity filtering (≤40%) and function-based redundancy removal to eliminate homologous or functionally overlapping proteins across species:
> * For Arath, MMSeqs2 removed 7,487 proteins, and 759 were excluded due to shared functions with Human, resulting in 5,025 final proteins (62%↓).
> * For Yeast, MMSeqs2 removed 1,033 proteins, and 1,026 shared the same function as Human (39%↓).
> * For Ecoli, MMSeqs2 removed 452 proteins, and 207 shared functions with Human (17%↓).
>
> We will include these results in Appendix C.3 of the revised manuscript to enhance the transparency and reproducibility of our dataset construction process.
>
>
> > W3: Compared with many existing PPI network datasets, the reviewers felt that the PPI network proposed in the manuscript lacked obvious innovation. Intuitively, existing PPI datasets also seem to be able to achieve the graph evaluations.
>
> **Response:** Thank you for pointing this out. We want to emphasize that existing PPI datasets are not well-suited for graph-level evaluation.
>
> **Prior datasets mostly contain incomplete network structures that does not support graph-level evaluation.** This is because most prior datasets are constructed by sampling a limited number of positive edges from larger PPI databases [3,4]. For example, the D-SCRIPT dataset [3] selects only 5,000 positive samples from STRING for cross-species analysis, leading to sparse and fragmented graphs.
>
>
> **PRING is built to preserve as much of the original interaction network as possible.** Rather than sampling isolated edges, we filter out redundant nodes from the full PPI network and retain the complete set of interactions among the remaining proteins. This results in more connected and biologically meaningful graphs. Moreover, PRING includes a large number of proteins across multiple species, enabling more comprehensive and realistic evaluation of PPI prediction models in both topological and functional contexts.
>
> We will make this distinction clearer in the revised manuscript to address potential concerns from readers.
>
>
> > W4: The dataset files are easy to obtain and have a clear format. It is recommended that the data format in each file be displayed more clearly.
>
> **Response:**
> Thank you for pointing this out. To improve clarity, we provide the detailed data format for each file below, using the intra-species (Human) setting as an example:
> |File Name|Description|
> |---|---|
> |`human.fasta`|FASTA file with entries in the format: `>uniprot_id` followed by amino acid sequence|
> |`human_train_ppi.txt`|Tab-separated file containing training PPI pairs: `uniprot_id1 uniprot_id2 label`|
> |`human_graph.pkl`|NetworkX graph object containing the full Human PPI network|
> |`Human_BFS_sampled_nodes.pkl`|A list of node sets corresponding to test subgraphs sampled via BFS from the full PPI graph|
>
> We thank this suggestion to help improve the usability of *PRING* dataset, and we will upload these additional information to our project repository after the review period.
>
> **Reference**
>
> [1] Szklarczyk, Damian, et al. "The STRING database in 2023: protein–protein association networks and functional enrichment analyses for any sequenced genome of interest." Nucleic acids research 51.D1 (2023): D638-D646.
>
> [2] Martha, Venkata-Swamy, et al. "Constructing a robust protein-protein interaction network by integrating multiple public databases." BMC bioinformatics 12.Suppl 10 (2011): S7.
>
> [3] Sledzieski, Samuel, et al. "D-SCRIPT translates genome to phenome with sequence-based, structure-aware, genome-scale predictions of protein-protein interactions." Cell Systems 12.10 (2021): 969-982.
>
> [4] Chen, Muhao, et al. "Multifaceted protein–protein interaction prediction based on Siamese residual RCNN." Bioinformatics 35.14 (2019): i305-i314.

---

> > ### Comment · Reviewer_bqYf · 2025-08-04
> >
> > Thanks to the author for his response to my comment. It clears up my confusion about the cross-species PPIs. And it explains why it didn't use the more complex method of merging PPI data. I think so far the manuscript meets the criteria for acceptance. I decide to keep my score.

---

> > > ### Author Response · Authors · 2025-08-04
> > > **Response to Reviewer bqYf**
> > >
> > > We are pleased that our clarifications addressed your concerns. We sincerely appreciate your positive review of our work, and your helpful suggestions have significantly contributed to the improvement of our manuscript's quality.

---

### Official Review · Reviewer_aprp · 2025-07-01

**Rating:** 4
**Confidence:** 3

**Summary:**

This paper introduces PRING, a benchmark designed to evaluate protein-protein interaction (PPI) prediction models at the graph level, as opposed to traditional pairwise evaluations. PRING includes a carefully curated multi-species dataset (Human, Arabidopsis, E. coli, Yeast) with rigorous redundancy reduction and leakage prevention protocols. The benchmark defines five evaluation tasks: two topology-oriented (intra-species and cross-species network reconstruction) and three function-oriented (complex pathway recovery, GO enrichment analysis, essential protein identification). Extensive experiments with 11 baseline models across four categories (sequence similarity, naive sequence, PLM-based, and structure-based) demonstrate that current models often fail to reconstruct biologically meaningful interaction networks, either topologically or functionally. PRING is released with full code and data for reproducibility.

**Dataset Code Accessibility:**

Yes

**Ethical Considerations:**

No, there are no or only very minor ethics concerns

**Final Justification:**

Given the rebuttal, I will maintain my positive rating.

**Limitations Weaknesses:**

1. The current structure-function evaluations are largely separate, without a unified metric or task linking them.

2. Cross-species transfer evaluation does not account for phylogenetic distance or functional annotation differences, which may introduce confounding effects.

3. While the paper observes that standard metrics like recall hurt topology preservation, the analysis remains shallow.

**Strengths Contributions:**

1. The work addresses a critical gap in the PPI prediction literature—namely, the lack of graph-level evaluation—and proposes a new standard for assessing models beyond binary classification.

2. The authors present a well-constructed, multi-species dataset with robust filtering strategies, which is both non-redundant and leakage-free, enhancing the validity of evaluations.

3. The benchmark includes both structural and functional evaluations, offering a holistic view of model capability across biologically relevant axes.

---

> ### Author Rebuttal · Authors · 2025-07-31
>
> We would like to express our sincere gratitude to the reviewer for the constructive comments and valuable suggestions. We are pleased that the reviewer appreciated various aspects of our work, including the critical research problem, well-constructed dataset, and holistic evaluations. Meanwhile, we have carefully considered the weaknesses raised and addressed them point by point below:
>
> > W1: The current structure-function evaluations are largely separate, without a unified metric or task linking them.
>
> **Response:** Thank you for pointing this out. First, We would like to clarify that the structure and function evaluations in *PRING* are not largely separate, as the function evaluations are inherently dependent on the underlying network structure. For example, the construction of protein complex pathways in the function-oriented evaluation relies heavily on the topology of the PPI network, since pathways are often inferred based on the connectivity of interaction edges. Therefore, the function evaluation serves not only to assess biological coherence and downstream applicability, but also indirectly reflects the structural quality of the constructed PPI network. We will make this point clearer in our revised manuscript.
>
> Additionally, designing a unified metric is more challenging, since structure evaluation typically assesses topological fidelity (e.g., degree distribution), while function evaluation emphasizes biological relevance, such as pathway construction or GO enrichment analysis. These two aspects are not easily comparable or integrated into a single quantitative score. We plan to explore this direction in our future work.
>
> > W2: Cross-species transfer evaluation does not account for phylogenetic distance or functional annotation differences, which may introduce confounding effects.
>
> **Response:** Thank you for the insightful suggestion. We respond to the two aspects raised, **phylogenetic distance** and **functional annotation differences**, respectively, below.
>
> **1) Phylogenetic Distance**: We agree that accounting for numerical phylogenetic distance can provide a deeper understanding of cross-species generalization. To investigate this, we conducted an ablation study by computing the **Pearson correlation between model performance and the estimated phylogenetic distance from Human**.
>
> Specifically, we set the intra-species (Human→Human) distance to 0, and adopt widely accepted phylogenetic estimates [1] for other species: Human→Arath (1.5 Gyr), Human→Yeast (1.2 Gyr), and Human→Ecoli (3.6 Gyr), which consist the distance vector $\mathbf{d}=[0,1.5,1.3,3.6]$. For a given model $M$ and performance metric $m$ (e.g., graph similarity (GS)), let $\mathbf{p}_m^M =[p{\scriptstyle\mathrm{Human}},p{\scriptstyle\mathrm{Arath}},p{\scriptstyle\mathrm{Yeast}},p{\scriptstyle\mathrm{Ecoli}}]$ denotes the model’s performance vector across the four species. We calculate the Pearson correlation r($\mathbf{d}$,$\mathbf{p}_m^{M}$) to quantify the relationship between phylogenetic distance and model performance. This analysis is conducted on four representative baselines, and are summarized in **Tab. 1**.
>
> **Tab. 1: Pearson Correlation Between Phylogenetic Distance and Graph-level Metrics**
>
> **a: BFS Sampling**
> |Method|r($\mathbf{d}$,$\mathbf{p}_{\text{GS}}$)|r($\mathbf{d}$,$\mathbf{p}_{\text{RD}}$)|r($\mathbf{d}$,$\mathbf{p}_{\text{Deg.}}$)|r($\mathbf{d}$,$\mathbf{p}_{\text{Clus.}}$)|r($\mathbf{d}$,$\mathbf{p}_{\text{Spectral}}$)|
> |---|:---:|:---:|:---:|:---:|:---:|
> |D‑SCRIPT|‑0.82|0.94|0.61|0.92|0.93|
> |PPITrans|‑0.98|0.80|0.94|0.56|1.00|
> |TUnA|‑0.97|0.97|0.96|0.94|0.92|
> |Struct2Graph|‑0.98|‑0.26|1.00|‑0.01|‑0.55|
>
> **b: DFS Sampling**
> |Method|r($\mathbf{d}$,$\mathbf{p}_{\text{GS}}$)|r($\mathbf{d}$,$\mathbf{p}_{\text{RD}}$)|r($\mathbf{d}$,$\mathbf{p}_{\text{Deg.}}$)|r($\mathbf{d}$,$\mathbf{p}_{\text{Clus.}}$)|r($\mathbf{d}$,$\mathbf{p}_{\text{Spectral}}$)|
> |---|:---:|:---:|:---:|:---:|:---:|
> |D‑SCRIPT|‑0.70|0.81|0.65|0.63|‑0.30|
> |PPITrans|‑0.90|0.89|0.53|0.81|0.79|
> |TUnA|‑0.98|0.98|0.49|0.70|0.64|
> |Struct2Graph|‑0.85|‑0.03|0.43|0.98|‑0.53|
>
> **c: Random Walk Sampling**
> |Method|r($\mathbf{d}$,$\mathbf{p}_{\text{GS}}$)|r($\mathbf{d}$,$\mathbf{p}_{\text{RD}}$)|r($\mathbf{d}$,$\mathbf{p}_{\text{Deg.}}$)|r($\mathbf{d}$,$\mathbf{p}_{\text{Clus.}}$)|r($\mathbf{d}$,$\mathbf{p}_{\text{Spectral}}$)|
> |---|:---:|:---:|:---:|:---:|:---:|
> |D‑SCRIPT|‑0.69|0.91|0.89|0.81|0.93|
> |PPITrans|‑0.91|0.86|0.87|0.77|0.95|
> |TUnA|‑0.96|0.95|0.92|0.86|0.97|
> |Struct2Graph|‑0.88|0.63|0.84|0.17|0.61|
>
> **Observations**: We find that **phylogenetic distance is strongly correlated with model performance degradation across species**.
> * **Graph Similarity (GS)**: For all models and sampling strategies, GS consistently exhibits a strong negative correlation with phylogenetic distance (correlations typically < -0.7). This indicates that as phylogenetic distance increases, GS decreases—i.e., model predictions become less similar to the ground-truth network.
> * **Other Metrics (RD, Deg., Clus., Spectral)**: Most other metrics show strong positive correlations with phylogenetic distance. This means that as the evolutionary gap widens, predicted networks increasingly diverge from true biological networks, reflected by higher RD, degree, clustering, or spectral values, all of which signal degraded performance.
>
> We really appreciate the reviewer's suggestion and find it meaningful to enhance the overall quality of our manuscript. We will include the numerical phylogenetic distance analysis in our revised manuscript to further improve the depth and clarity of our cross-species evaluation.
>
> **2) Functional Annotation Difference**: In our current study, we do not incorporate function-oriented evaluations in our cross-species setting. The GO enrichment (function annotation) analysis is conducted only within the intra-species (Human) context to better interpret biological patterns. We will make this point clearer in our revised manuscript.
>
>
> > W3: While the paper observes that standard metrics like recall hurt topology preservation, the analysis remains shallow.
>
> **Response:** Thank you for the suggestion on improving the presentation of our experimental analysis. We have included additional analysis on the correlation between standard metrics and network topology in Appendix G.1.3, but only present the reall and precision in the main paper. To address the reviewer's concern more directly, we summarize and clarify the key observations regarding correlation between binary classification scores and topological metrics below:
>
> * **Recall often distorts the network topology:** Recall measures the proportion of true positives correctly identified. To achieve higher recall, the model typically predicts more edges to capture as many true interactions as possible, even at the cost of including false positives. This leads to a denser predicted graph, which deviates from the sparsity characteristic of real PPI networks, ultimately hurting the preservation of the original topological structure.
> * **Precision tends to preserve network topology:** Precision measures the proportion of correct positive predictions. Improving precision requires minimizing false positives, which makes the model more conservative in predicting edges. As a result, the predicted graph becomes sparser and more aligned with the intrinsic sparsity of real PPI networks, thereby helping to preserve the underlying network topology.
> * **Composite metrics like F1-score and AUPR may hide topological differences:** Since these metrics balance both precision and recall, they may yield high scores even when the predicted network substantially deviates from the true topology. As a result, their correlation with topological fidelity tends to be weak, potentially masking structural distortions in the predicted network.
>
> We will include the above detailed analysis in our revised manuscript.
>
> **Reference**
>
> [1] Kumar, Sudhir, et al. "TimeTree 5: an expanded resource for species divergence times." Molecular biology and evolution 39.8 (2022): msac174.

---

> > ### Comment · Reviewer_aprp · 2025-08-04
> >
> > Thanks for the detailed rebuttal, I will maintain my positive rating on this paper.

---

> > > ### Author Response · Authors · 2025-08-04
> > > **Response to Reviewer aprp**
> > >
> > > We sincerely thank the reviewer for acknowledging our rebuttal and for a positive rating of our work. Your thoughtful feedback and suggestions have been instrumental in strengthening the quality of this manuscript.

---

### Official Review · Reviewer_vu3j · 2025-07-02

**Rating:** 5
**Confidence:** 3

**Summary:**

This paper creates a new protein-protein interaction graph dataset that includes multiple species and proposes new benchmarks to assess representative methods from graph perspective on topology-oriented tasks including intra and cross-species PPI network datasets and function oriented tasks including protein complex pathway prediction, GO module analysis, and essential protein justification to see if a model can preserve PPI network properties.

**Dataset Code Accessibility:**

Partly

**Dataset Code Comments:**

I didn't run the code by myself. But the repo is well-structured and the instructions are very clear. Both datasets and benchmark code can be found in the repo.

**Ethical Considerations:**

No, there are no or only very minor ethics concerns

**Final Justification:**

Given the thorough experiments and the authors' detailed explanation during rebuttal, I'd like to keep positive.

**Limitations Weaknesses:**

1. Some background isn't explained well, hindering researchers from understanding why the four datasets are sampled.
2. It seems like the authors forgot to explain the step between the protein filter and the similarity check in Figure 2. I don't understand how the source database and PRING dataset are connected.

**Strengths Contributions:**

1. The paper is well-written. The research problem is well-motivated, and the evaluation strategies are well-formulated.
2. The evaluation is thorough. The authors use three subgraph sampling methods to generate subgraphs and
3. The paper has insightful findings that current PPI models are not able to preserve the structural information, so that not able to support real-world biological applications, urging more research on this matter.

---

> ### Author Rebuttal · Authors · 2025-07-31
>
> We would like to express our sincere gratitude to the reviewer for the constructive comments and valuable suggestions. We are pleased that the reviewer appreciated various aspects of our work, including the well-motivated research problem, well-formulated evaluation strategies, thorough evaluations, and insightful findings. Meanwhile, we have carefully considered the weaknesses raised and addressed them point by point below:
>
> > W1: Some background isn't explained well, hindering researchers from understanding why the four datasets are sampled.
>
> **Response:** Thank you for pointing this out. We select UniProt, Reactome, IntAct, and STRING as our primary source databases based on their high data quality, broad coverage, and widely adoption in the community. The reasons are as follows:
>
> * **These four selected datasets offer high quality and broad coverage.** UniProt, Reactome, and IntAct primarily consist of experimentally validated protein-protein interactions (PPIs), which ensures the reliability of the data. For instance, IntAct provides manually curated interactions from the literature, supported by specific experimental evidence such as yeast two-hybrid. STRING, the largest and most comprehensive PPI resource, integrates PPIs from diverse sources, including experimental data, co-expression, text mining, and computational predictions. It spans more than 5,000 organisms and provides interaction confidence scores. Therefore, collecting PPIs from these four datasets not only ensures data quality but also emphasizes data diversity and coverage, providing a critical foundation for constructing our final *PRING* dataset.
> * **Other PPI databases either substantially overlap with the selected sources or suffer from limited coverage and lower data quality.** Databases such as BioGRID [1], MINT [2], and DIP [3] are mostly integrated into STRING and IntAct. Including them separately would introduce redundancy and risk duplicate entries. Other datasets have limited interaction coverage. For example, HuRI [4] mainly focus on the human proteome and focus on certain interaction types, such as membrane proteins. In addition, some resources such as PrePPI [5] rely heavily on computational predictions without experimental validation, which increases the risk of false positives and reduces overall reliability.
> * **These four dataset are also widely adopted in the community.** Prior studies [6,7] have consistently used UniProt, Reactome, IntAct, and STRING as core sources when building PPI datasets or knowledge graphs. Following this well-established practice not only ensures reproducibility and comparability with previous work, but also provides a reliable basis for downstream analysis.
>
> We will make this motivation more explicit in our revised manuscript.
>
>
> > W2: It seems like the authors forgot to explain the step between the protein filter and the similarity check in Figure 2. I don't understand how the source database and PRING dataset are connected.
>
> **Response:** Thank you for pointing this out. Here, we provide a more detailed explanation to elaborate the two steps in Figure 2. After collecting high-quality PPIs from four source databases, we apply two critical filtering steps before constructing the final *PRING* dataset.
>
> 1. **Protein Filter**: We use SwissProt to retain only proteins with well-annotated functions and apply NCBI Taxonomy identifiers to ensure that each protein is correctly assigned to one of the four target species. This step removes proteins with unknown functions or ambiguous species information.
> 2. **Similarity Check**: We reduce data redundancy by performing sequence clustering using MMSeqs2 [8], retaining only protein pairs with sequence identity ≤ 40%, following prior works [9,10]. In addition, we remove proteins that share identical function IDs across species to ensure non-redundancy further.
>
> We appreciate the reviewer’s suggestion for improving the clarity of our manuscript. We will describe these steps more clearly in our revised manuscript.
>
>
> **Reference**
>
> [1] Stark, Chris, et al. "BioGRID: a general repository for interaction datasets." Nucleic acids research 34.suppl_1 (2006): D535-D539.
>
> [2] Chatr-Aryamontri, Andrew, et al. "MINT: the Molecular INTeraction database." Nucleic acids research 35.suppl_1 (2007): D572-D574.
>
> [3] Xenarios, Ioannis, et al. "DIP: the database of interacting proteins." Nucleic acids research 28.1 (2000): 289-291.
>
> [4] Luck, Katja, et al. "A reference map of the human binary protein interactome." Nature 580.7803 (2020): 402-408.
>
> [5] Zhang, Qiangfeng Cliff, et al. "PrePPI: a structure-informed database of protein–protein interactions." Nucleic acids research 41.D1 (2012): D828-D833.
>
> [6] Doğan, Tunca, et al. "CROssBAR: comprehensive resource of biomedical relations with knowledge graph representations." Nucleic Acids Research 49.16 (2021): e96-e96.
>
> [7] Chandak, Payal, Kexin Huang, and Marinka Zitnik. "Building a knowledge graph to enable precision medicine." Scientific Data 10.1 (2023): 67.
>
> [8] Steinegger, Martin, and Johannes Söding. "MMseqs2 enables sensitive protein sequence searching for the analysis of massive data sets." Nature biotechnology 35.11 (2017): 1026-1028.
>
> [9] Bernett, Judith, David B. Blumenthal, and Markus List. "Cracking the black box of deep sequence-based protein–protein interaction prediction." Briefings in Bioinformatics 25.2 (2024).
>
> [10] Sledzieski, Samuel, et al. "D-SCRIPT translates genome to phenome with sequence-based, structure-aware, genome-scale predictions of protein-protein interactions." Cell Systems 12.10 (2021): 969-982.

---

> > ### Comment · Area_Chair_Qyvo · 2025-08-06
> >
> > Dear reviewer, could you please check the rebuttal and respond to it?

---

> > ### Comment · Reviewer_vu3j · 2025-08-07
> >
> > Thanks for the authors' response. I have read their rebuttals and remain positive on this paper.

---

> > > ### Author Response · Authors · 2025-08-07
> > > **Response to Reviewer vu3j**
> > >
> > > Thank you for acknowledging our rebuttal and for your positive comments on our work. We appreciate your valuable suggestions, which have helped us further improve the presentation of our work.

---

### Official Review · Reviewer_T2pD · 2025-07-04

**Rating:** 5
**Confidence:** 2

**Summary:**

The paper's central thesis is that existing benchmarks which evaluate pairwise interactions between proteins are insufficient to asses the model's real-world utility since they ignore the how individual predictions contribute to cohesive PPI networks. To overcome this limitaiton, they introduces a benchmark designed to assess protein protein interactions at the graph level. The benchmark spans four species and consists of 21K proteins and 187K pairs, and studies both the topological and functional characteristics of the predictions. The paper contains an exhaustive evaluation of 11 baselines and reveal that strongest models perform poorly on both graph construction as well as predicting its function.

**Dataset Code Accessibility:**

Yes

**Ethical Considerations:**

No, there are no or only very minor ethics concerns

**Final Justification:**

The authors ran additional experiments during the rebuttal which strengths the paper. I am happy with to keep my positive rating.

**Limitations Weaknesses:**

- The paper used a 1:1 positive:negative ratio for training. While this makes sense to balance the training dataset, it likely biases the model to predict denser graphs despite ground truth likely having higher sparsity. It'd be informative to ablate this choice with different ratios.
- On a related note, discussion about how the probabilities are thresholded and how that affects precision-recall might also be useful.
- For the generalization experiments, transfer is studied only along one direction (Human -> Other species). Do we expect similar conclusions for other settings?

**Strengths Contributions:**

- A benchmark to measure the ability to construct the entire network instead of just edges, as well as predicting both the topological and functional aspects is important and a significant contribution.
- The authors have taken care curating the benchmark by removing redundancy, preventing data leakage between train and test datasets.
- The analysis is comprehensive -- evaluating 11 baselines spanning different style of methods (sequence similarity based model, PLMs, Structgure based models etc).

---

> ### Author Rebuttal · Authors · 2025-07-31
>
> We sincerely thank the reviewer for the constructive feedback and for acknowledging our efforts in developing a rigorous and meaningful benchmark, especially our efforts to reduce redundancy, prevent data leakage, and conduct comprehensive baseline evaluations. Meanwhile, we have carefully considered the weaknesses raised and addressed each point below.
>
> > W1: The paper used a 1:1 positive:negative ratio for training. While this makes sense to balance the training dataset, it likely biases the model to predict denser graphs despite ground truth likely having higher sparsity. It'd be informative to ablate this choice with different ratios.
>
> **Response:** Thank you for the suggestion. To address this, we conduct additional experiments treating the positive-to-negative ratio as a tunable hyperparameter to examine its effect on the density of predicted graphs. Due to time constraints during the rebuttal period, we focus on two settings: **1:5** and **1:10**, and limit our evaluation to intra-species (Human) topology-oriented tasks. We evaluate four representative baselines spanning three categories: sequence-based (D-SCRIPT), PLM-based (PPITrans, TUnA), and structure-based (Struct2Graph). Notably, SPRINT was excluded, as it does not require negative samples for training. The results are summarized in **Tab. 1**.
>
> **Tab. 1: Ablation Study on Positive-to-Negative Ratio**
> Relative Density (RD) measures how much the predicted network overpredicts (RD > 1) or underpredicts (RD < 1) edges compared to the ground-truth network’s edge density.
>
> **a: BFS Sampling**
> |||Ratio 1:1||||
> |---|:---:|:---:|:---:|:---:|:---:|
> |Method|GS$\uparrow$|RD$\rightarrow$|Deg.$\downarrow$|Clus.$\downarrow$|Spectral$\downarrow$|
> |D‑SCRIPT|0.215|1.13|15.4|18.8|29.7|
> |PPITrans|0.362|3.39|52.2|29.1|26.3|
> |TUnA|0.342|2.99|47.8|30.2|22.9|
> |Struct2Graph|0.270|5.58|95.1|50.5|39.2|
> |||**Ratio 1:5**||||
> |D‑SCRIPT|0.032|0.0705|94.2|46.6|83.2|
> |PPITrans|0.393|1.17|8.12|7.37|10.9|
> |TUnA|0.346|1.14|10.7|9.06|8.77|
> |Struct2Graph|0.236|4.11|62.6|35.2|25.6|
> |||**Ratio 1:10**||||
> |D‑SCRIPT|0.035|0.04|116|54.1|97.2|
> |PPITrans|0.374|0.692|11.3|7.23|16.4|
> |TUnA|0.314|0.579|14.4|9.58|17.3|
> |Struct2Graph|0|5.71E‑05|146|65.8|117|
>
> **b: DFS Sampling**
> |||Ratio 1:1||||
> |---|:---:|:---:|:---:|:---:|:---:|
> |Method|GS$\uparrow$|RD$\rightarrow$|Deg.$\downarrow$|Clus.$\downarrow$|Spectral$\downarrow$|
> |D‑SCRIPT|0.123|6.40|302|237|67.0|
> |PPITrans|0.314|4.54|418|270|46.6|
> |TUnA|0.289|4.44|416|272|46.0|
> |Struct2Graph|0.132|10.8|508|321|67.6|
> |||**Ratio 1:5**||||
> |D‑SCRIPT|0.0649|0.0545|518|137|136|
> |PPITrans|0.433|2.19|213|153|28.3|
> |TUnA|0.417|1.49|127|85.6|19.7|
> |Struct2Graph|0.118|7.29|430|282|50.9|
> |||**Ratio 1:10**||||
> |D‑SCRIPT|0.089|0.27|413|112|107|
> |PPITrans|0.520|1.19|100|40|38|
> |TUnA|0.414|0.933|114|21.9|38.9|
> |Struct2Graph|1.74E‑05|3.50E‑05|593|154|145|
>
> **c: Random Walk Sampling**
> |||Ratio 1:1||||
> |---|:---:|:---:|:---:|:---:|:---:|
> |Method|GS$\uparrow$|RD$\rightarrow$|Deg.$\downarrow$|Clus.$\downarrow$|Spectral$\downarrow$|
> |D‑SCRIPT|0.209|5.82|134|64.8|52.0|
> |PPITrans|0.449|2.57|71.9|46.1|19.6|
> |TUnA|0.450|2.23|55.4|42.5|13.7|
> |Struct2Graph|0.237|5.83|155|103|38.3|
> |||**Ratio 1:5**|||
> |D‑SCRIPT|0.135|0.334|83.2|51.9|55.6|
> |PPITrans|0.564|1.27|10.8|10.8|12.3|
> |TUnA|0.463|0.764|15.8|9.58|13.2|
> |Struct2Graph|0.216|3.44|87.7|72|18.4|
> |||**Ratio 1:10**|||
> |D‑SCRIPT|0.01|0.01|188|133|134|
> |PPITrans|0.547|0.808|23.7|14.9|25|
> |TUnA|0.425|0.539|38.0|27.6|29.7|
> |Struct2Graph|0|5.71E‑05|197|135|140|
>
> **Observations**:
> 1. **1:5 ratio improves alignment with real PPI sparsity.** Increasing the negative sampling ratio to 1:5 leads to a reduction in the relative density of predicted graphs across BFS, DFS, and Random Walk strategies for models like PPITrans, TUnA, and Struct2Graph. This shift indicates better alignment with the inherent sparsity in real-world PPI networks, while also improving other graph-level metrics.
> 2. **1:10 ratio is too unbalanced and harms performance.** Most models become overly conservative, rarely predicting positive interactions under this setting, with densities falling below 1 and performance degradation.
>
> We thank the reviewer again for this valuable suggestion, which we find useful to improve the model's performance by tuning the positive to negative ratio. We believe the new findings enhance the comprehensiveness of our work. Additionally, we acknowledge that a more detailed study of the positive-to-negative ratio could yield further insights, and that selecting an optimal value for this hyperparameter remains an open question for different tasks and datasets. We will include these new findings in the revised manuscript and leave the broader investigation for future work.
>
>
> > W2: On a related note, discussion about how the probabilities are thresholded and how that affects precision-recall might also be useful.
>
> **Response:** Thank you for pointing this out. In our original experiments, **we adopted the default threshold of 0.5 without any hyperparameter tuning for binary classification to ensure a fair and unbiased comparison**, following the settings used in prior works [1].
>
> To further investigate this problem, we perform an ablation study by varying the threshold from **0.2** to **0.8** and evaluating its impact on precision and recall across four baseline models. The results are presented in **Tab. 2**.
>
> **Tab. 2: Effect of Probability Thresholds on Precision and Recall Across Four Models**
> ||PIPR||D‑SCRIPT||PPITrans||Struct2Graph||
> |:--:|:--:|:--:|:--:|:--:|:--:|:--:|:--:|:--:|
> |Threshold|Prec.|Rec.|Prec.|Rec.|Prec.|Rec.|Prec.|Rec.|
> |**0.2**|0.552|0.634|0.570|0.135|0.627|0.821|0.502|0.937|
> |**0.3**|0.554|0.615|0.582|0.115|0.651|0.769|0.504|0.900|
> |**0.5**|0.557|0.581|0.597|0.104|0.658|0.738|0.511|0.696|
> |**0.7**|0.561|0.550|0.764|0.067|0.753|0.535|0.532|0.007|
> |**0.8**|0.562|0.530|0.780|0.053|0.798|0.438|0.532|0.007|
>
> **Observations**: Struct2Graph and D-SCRIPT are highly sensitive to thresholding, with recall dropping sharply, indicating that a large portion of their prediction scores have low confidence and are clustered around 0.5. In contrast, PIPR and PPITrans exhibit more stable trends: recall gradually decreases while precision increases as the threshold rises, suggesting a trade-off between the two metrics and a limited ability to maintain a strong balance.
>
> We will highlight our default threshold setting and include the new analysis in our revised manuscript to improve the presentation.
>
> > W3: For the generalization experiments, transfer is studied only along one direction (Human -> Other species). Do we expect similar conclusions for other settings?
>
> **Response:** Thank you for the suggestion. Following your instruction, we perform new experiments on botany-to-fungi (Arath → Yeast) and botany-to-bacteria (Arath → Ecoli) transfer learning. We followed the same data construction pipeline described in Sec. 3.1 of the main paper to avoid data leakage. Due to time constraints during the rebuttal period, we focused on the BFS sampling strategy and evaluated four representative models. The results are presented in **Tab. 3**, and they confirm our preivous experimental observation: **cross-species generalization becomes more challenging with greater evolutionary divergence**.
>
> **Tab. 3: Alternative Cross-species Transfer**
>
> **a: Arath→Arath**
> |Method|GS$\uparrow$|RD$\rightarrow$|Deg.$\downarrow$|Clus.$\downarrow$|Spectral$\downarrow$|
> |---|:---:|:---:|:---:|:---:|:---:|
> |D‑SCRIPT|0.407|2.24|38.4|11.6|19.0|
> |PPITrans|0.470|2.96|43.7|31.4|17.8|
> |TUnA|0.548|1.54|22.6|19.1|14.4|
> |Struct2Graph|0.352|2.59|67.0|70.5|16.3|
>
> **b: Arath→Yeast**
> |Method|GS$\uparrow$|RD$\rightarrow$|Deg.$\downarrow$|Clus.$\downarrow$|Spectral$\downarrow$|
> |---|:---:|:---:|:---:|:---:|:---:|
> |D‑SCRIPT|0.288|3.68|52.3|25.0|28.9|
> |PPITrans|0.376|4.68|78.0|50.8|35.8|
> |TUnA|0.378|3.87|58.5|42.9|27.0|
> |Struct2Graph|0.266|4.84|92.9|62.1|30.6|
>
> **c: Arath→Ecoli**
> |Method|GS$\uparrow$|RD$\rightarrow$|Deg.$\downarrow$|Clus.$\downarrow$|Spectral$\downarrow$|
> |---|:---:|:---:|:---:|:---:|:---:|
> |D‑SCRIPT|0.243|6.11|121|46.1|44.1|
> |PPITrans|0.312|5.14|138|51.4|44.1|
> |TUnA|0.302|5.39|129|49.9|42.1|
> |Struct2Graph|0.218|5.53|145|61.0|38.8|
>
> **Observations**: Specifically, compared to the intra-species setting (Arath → Arath), the graph similarity (GS) scores for Arath → Yeast and Arath → Ecoli decreased by 26% and 39%, respectively. Other graph-level metrics observe performance degradation as well, indicating the performance of the transfer learning decreases with the increased evolutionary distances.
>
> **Reason for not including other transfer directions in the original manuscript.** Methodologically, our original setting follows prior works [1, 2] and is widely adopted to assess cross-species generalization. Biologically, Human is the most extensively studied organism, with the largest number of annotated proteins in UniProt (20,421 with verified functions), compared to Arath (16,397), Yeast (6,733), and Ecoli (6,067). Thus, using Human as the source species reflects both data richness and practical relevance in biomedical research.
>
> We thank the reviewer for this suggestion, which has helped improve the completeness of our study. We will include the new transfer learning experiments in the revised manuscript.
>
> **Reference**
>
> [1] Yang, Sen, et al. "Exploring the knowledge of an outstanding protein to protein interaction transformer." IEEE/ACM Transactions on Computational Biology and Bioinformatics 21.5 (2024): 1287-1298.
>
> [2] Sledzieski, Samuel, et al. "D-SCRIPT translates genome to phenome with sequence-based, structure-aware, genome-scale predictions of protein-protein interactions." Cell Systems 12.10 (2021): 969-982.

---

> > ### Comment · Area_Chair_Qyvo · 2025-08-06
> >
> > Dear reviewer, could you please check the rebuttal and respond to it?

---

> > ### Comment · Reviewer_T2pD · 2025-08-08
> > **Thanks for the rebuttal**
> >
> > Thank you for running additional experiments during the rebuttal. I agree with the observations and recommend the authors to discuss these new experiments in the final submission. I am happy with my rating.

---

> > > ### Author Response · Authors · 2025-08-08
> > > **Response to Reviewer T2pD**
> > >
> > > Thank you for recognizing our additional experiments and agreeing with our observations. We sincerely appreciate your positive review and insightful comments. We will include the additional experiments and discussions in the revised manuscript to further enhance the comprehensiveness of our work.

---

### Note · Authors · 2025-08-13

Dear Reviewers, ACs, SACs, and PCs,

We thank all reviewers for their thoughtful comments and the AC for coordinating the rebuttal process. Across the reviewers, several key strengths are consistently recognized:
* **[Reviewer T2pD, vu3j, aprp, and bqYf]** A novel and important **graph-level** PPI benchmark beyond pairwise classification to jointly assess topological and functional aspects.
* **[Reviewer T2pD, aprp, and bqYf]** **Rigorous curation** with redundancy removal and leakage prevention across multiple species.
* **[Reviewer T2pD, vu3j, and bqYf]** **Thorough evaluation** over 11 baselines with released code and data.
* **[Reviewer vu3j and bqYf]** **Insightful findings** reveal that current PPI models struggle to preserve structural information and real-world biological applicationw, and calling for further research.

During the rebuttal period, we address the sugestions, questions, and weaknesses raised by each reviewer as follows:

* **[Reviewer T2pD]** We perform additional ablations using 1:5 and 1:10 positive-to-negative ratios, explore different threshold ranges to analyze the precision–recall trade-off, and extend cross-species transfers (Arath→Yeast/E. coli) to cover new scenarios.
* **[Reviewer vu3j]** We explain the reason for selecting four source databases and provide a more detailed description of Protein Filter and Similarity Check in Figure 2.
* **[Reviewer aprp]** We clarify the relationship between topology- and function-oriented tasks, quantify performance against TimeTree-based phylogenetic distance, and analyze in detail how binary metrics (Precision, Recall, F1, AUPR) affect topology fidelity.
* **[Reviewer bqYf]** We justify the rationality of our data merging strategy, provide additional statistics of our *PRING*, and explain why prior PPI datasets are unsuitable for graph-level evaluations.

We appreciate that all reviewers acknowledge our clarifications and maintain their positive scores. Their constructive feedback has been invaluable, helping us improve the quality of our work. We believe these additions further strengthen *PRING*, and we will incorporate them into our revised manuscript.

Thank you for your attention and support.

Sincerely,

Paper 763 Authors

---

### Decision · Program_Chairs · 2025-09-18

**Decision:**

Accept (poster)

**Comment:**

This work developed a multi-species protein-protein interaction (PPI) dataset focused on evaluating PPI predictions from the graph perspective rather than the traditional pairwise interaction (edge) perspective. The dataset was used to establish two new evaluation paradigms: (1) network topology-oriented tasks and (2) function-oriented tasks. The evaluation identified the limitations of current PPI prediction methods in recovering the structural and functional properties of PPI networks.

The following strengths of the work were identified during the review process: (1) the benchmark evaluated the ability of reconstructing entire PPI networks instead of only edges and predicting both topological and functional properties of PPI networks; (2) data redundancy was removed to prevent data leakage; (3) 11 different baseline methods were evaluated; (4) different sub-graph sampling methods were used in the evaluation; (5) the benchmark identified the limitation of the existing PPI prediction methods in predicting network topology and function; and (6) both structural and functional evaluations were included in the benchmark.

The following weaknesses (or suggestions for improvement) were identified (or made) during the initial review: (1) adding the ablation study of the positive and negative data ratio in the training; (2) testing different probability thresholds of making binary decision predictions; (3) testing the generalization ability between more species; (4) a lack of the background information about the four sampled datasets and about the relationship between the source databases and PRING dataset; (5) a lack of unified metric to combine the structural and functional evaluation; (6) cross-species transfer evaluation did not account for phylogenetic distance or functional annotation differences; (7) no sophisticated method was proposed to merge the datasets; and (8) a lack of detailed statistical analysis of the PPI networks.

During the rebuttal process, Points 1, 2, 3, 4, 6, 8 above were all well addressed by either new experiments/data analysis or text revision.  As for Points 5 and 7, due to the difficulty of creating a reasonable unified metric to assess both structural and functional evaluation or proposing a novel workable dataset merging method, it is reasonable for the authors to postpone it to the future work.

Overall, this work addresses a significant gap in evaluating PPI networks and is a useful resource for developing and benchmarking PPI predictions. All concerns raised by the reviewers were addressed well during the rebuttal process. Therefore, I recommend the acceptance of this work.